# DUO: No Compromise to Accuracy Degradation

**Jinda Jia**[1]*, **Cong Xie**[2]*†, **Fanjiang Ye**[1], **Hao Feng**[1],
**Hanlin Lu**[2], **Daoce Wang**[1], **Haibin Lin**[2], **Zhi Zhang**[2] **and Xin Liu**[2]
[1]Indiana University, [2]ByteDance Inc.
{jindjia, fanjye, haofeng, daocwang}@iu.edu
{cong.xie, hanlin.lu, haibin.lin, zhangzhi.joshua, liuxin.ai}@bytedance.com

## Abstract

Distributed training often suffers from high communication overhead due to large-scale gradient synchronization. Although gradient compression—particularly at 4-bit or even lower precision—significantly reduces transfer volume, it typically results in sacrifice in precision and degradation of the final model accuracy.

In this work, we introduce **DUO** (*Double Update Overlap*), a distributed training framework designed to mitigate accuracy degradation caused by gradient compression without introducing additional overhead. DUO achieves this by inserting an additional high-precision gradient synchronization step into a previously computation-only phase, so that its communication is fully hidden by computation.

We provide a comprehensive theoretical proof of convergence for DUO and validate its effectiveness through extensive pre-training experiments on GPT models. Our results indicate that DUO effectively restores accuracy when using 4-bit gradient compression, achieving performance comparable to uncompressed training. Remarkably, DUO maintains minimal accuracy degradation even under extreme compression scenarios, including **1-bit** gradients or **complete omission** of the low-precision gradient communication step (0-bit transmission).

## 1 Introduction

In recent years, transformer-based language models [3, 10, 25, 26] have demonstrated superior capabilities in learning and predicting sequence tokens based on contextual information. Thanks to the attention mechanism, traditional NLP tasks such as question answering, language translation and sentiment analysis have achieved remarkable improvements. Moreover, leveraging the attention mechanism inherent to transformers, researchers have extended their applications to multimodal domains, including image interpretation and video generation. Empirical evidence indicates that larger transformer-based models can provide more powerful generative capability and consequently improve output accuracy.

As model sizes continue to grow, distributed training has become essential for scaling deep learning, enabling the use of many GPUs to handle large datasets and complex architectures. However, it also introduces frequent GPU-to-GPU communication for synchronizing gradients and model parameters. These communications must complete before the next training step can begin, creating dependencies that stall GPU computation, thereby reducing overall throughput. As a result, the efficiency gains from multi-GPU parallelism are often undermined by communication overhead.

Recent work has also explored optimizer-level strategies to improve stability and reduce communication [5, 6, 7]; however, these approaches are orthogonal to ours, as they modify the optimizer and require retraining or retuning, while DUO preserves the existing training configuration and can be plugged in without modification.

---

*Equal Contribution.
†Corresponding author.

39th Conference on Neural Information Processing Systems (NeurIPS 2025).

Compression is a fundamental technique for reducing data size and has been widely applied in distributed training to alleviate both communication and memory overhead. Existing approaches include lossless methods (e.g., [2, 11, 34]) that preserve exact information, as well as lossy methods (e.g., [24, 27, 29, 32, 33]), which trade precision for efficiency. Among them, quantization has become the most widely used strategy in large-scale LLM training due to its low latency, compatibility with existing pipelines, and ease of integration. Recent work has successfully applied quantization to reduce communication overhead, such as ZeRO++ [28], SPD4Bit [14], QSDP [16] and FP8 training [18].

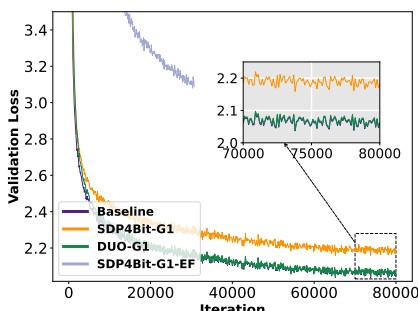

Figure 1: Training validation loss for GPT-350M

However, despite its efficiency, aggressive quantization often harms model accuracy—particularly during pre-training. Prior empirical studies [18, 16, 14, 28] show that gradients are highly sensitive to precision loss, making gradient compression especially challenging. While methods such as error feedback [15, 13, 21] aim to mitigate quantization error and recover lost accuracy, they perform poorly when applied to large language models (LLMs) pre-training under aggressive compression (e.g., 1-bit precision). As shown in Figure 1, the variant *SDP4Bit-G1-EF*—which uses 1-bit gradient compression with error feedback enabled—exhibits a significant accuracy gap relative to the baseline and fails to converge. In contrast, DUO achieves accuracy comparable to the baseline.

These observations reveal a fundamental trade-off between communication efficiency and model accuracy, *which raises an important question: can we compress gradients without sacrificing accuracy or throughput?* In this work, we answer this question affirmatively by proposing DUO, a high-accuracy communication-efficient training framework.

**Our contributions are as follows:**

- We introduce the Fast-Slow Reduction algorithm, designed to mitigate accuracy degradation due to gradient compression.
- We propose DUO, a communication-efficient training framework that integrates the Fast-Slow algorithm with system-level design to achieve high accuracy with negligible overhead.
- We establish a theoretical guarantee for the convergence of SGD with Fast-Slow algorithm, demonstrating that it maintains the same convergence rate as standard SGD.
- We integrate DUO into Megatron-LM and conduct extensive experiments to evaluate both its accuracy and efficiency. Results show that DUO achieves accuracy comparable to the uncompressed baseline, even under extreme communication compression settings (1-bit and 0-bit).

## 2 Background and Motivation

Megatron-LM [22] is a widely used framework for training large language models (LLMs). It employs three key strategies—Sharded Data Parallelism [36], gradient accumulation, and communication overlapping—all of which are closely related to the design of our proposed DUO framework. To provide context, we briefly introduce these strategies in the following sections.

### 2.1 Sharded Data Parallelism

Training LLMs such as GPT [1] and LLaMA [25] necessitates distributed training strategies to handle their extensive parameter sizes. Unlike naive Data Parallelism, Sharded Data Parallelism (ShardedDP) partitions high-precision optimizer states across all GPUs, which reduces the memory usage of the optimizer states by a factor of $n$, where $n$ represents the number of GPUs. As a result, ShardedDP changes the Naive Data Parallel communication pattern from AllReduce to a ReduceScatter plus an AllGather, as we summarized in the following equations:

$$w = \text{AllGather}\big(w[1], \dots, w[n]\big) \tag{1}$$

$$g_p = \nabla_w \mathcal{L}(\theta; D_p) \tag{2}$$

$$\bar{g}[p] = \text{ReduceScatter}(g_1, \dots, g_n) \tag{3}$$

$$w[p] \leftarrow \text{OptimizerStep}\big(w[p], \bar{g}[p]\big) \tag{4}$$

At the beginning of each training iteration, each worker $p$ holds its respective partition of the parameters $w[p]$ and the corresponding optimizer states. The training iteration proceeds as follows:

(1) An *AllGather* [8] operation synchronizes the model parameters across all workers. After this synchronization, each worker holds a full replica of the model parameters $w$.

(2) Each worker computes gradients $g_p$ based on its subset of data $D_p$.

(3) Gradients are synchronized using a *ReduceScatter* [9] operation. This results in each worker $p$ receiving a partition $\bar{g}[p]$ of the averaged gradients. In this paper, our main contribution is to optimize this ReduceScatter operation for speed without sacrificing accuracy.

(4) Each worker performs an optimizer step to update its local parameter partition $w[p]$ using the received gradient partition $\bar{g}[p]$, as well as the corresponding optimizer states.

Table 1: Notations of Sharded Data Parallelism.

| $p$ | index of worker or the corresponding shard/partition | $t$ | index of iteration |
|---|---|---|---|
| $w$ | parameter weights | $w_t$ | weights in iteration $t$ |
| $w[p]$ | $p$th shard of weights | $g$ | gradients |
| $g_{t,p}$ | gradient produced by worker $p$ in iteration $t$ | $\bar{g}$ | averaged gradients |
| $\tilde{g}$ | compressed gradients | $\bar{g}[p]$ | $p$th shard of averaged gradients |

## 2.2 Gradient Accumulation

Gradient accumulation is widely used for training LLMs with large batch sizes. Pre-training of LLMs typically requires massive batch sizes to improve training stability and efficiency. However, computing on the full batch at once leads to excessive GPU memory consumption.

To address this, gradient accumulation divides a large batch into multiple micro-batches. Each micro-batch is processed sequentially, with gradients locally accumulated in memory. Gradient synchronization and weight updates occur only after all micro-batches within a batch have been processed, reducing the frequency of communication-intensive synchronization.

As shown in Figure 2, gradient accumulation reduces communication frequency but introduces communication-idle gaps between synchronization steps. In this paper, DUO exploits these idle periods to overlap high-precision gradients communication that improve parameter and optimizer state accuracy.

## 2.3 Communication Overhead Optimization Strategy

As illustrated in the first diagram of Figure 2 and discussed in Section 2.1, Naive Sharded Data Parallelism requires two communication operations per iteration: an AllGather of weights and a ReduceScatter of gradients. During these communication phases, the GPU must wait for the communication to complete, remaining mostly idle and leading to significant computational waste during training.

Overlapping and compression [30, 33, 17, 20, 31, 4] are two commonly used methods for reducing communication overhead. We introduce these strategies in a progressive manner.

**Computation-Communication Overlapping.** To alleviate communication overhead, previous works [19, 36, 4, 22] have explored overlapping strategies to better utilize computation-heavy phases. The key idea is that gradient and weight communications need not be performed all at once; instead, they can be partitioned into multiple steps and overlapped with nearby computations.

As shown in second diagram of Figure 2, overlapping strategies divide computation into buckets. During the forward pass, the system performs computation on the current bucket while concurrently gathering weights for the next one. Similarly, during the backward pass, gradient computation for the current bucket overlaps with the gradient reduction of previous buckets.

However, modern GPUs offer significantly higher compute throughput than network bandwidth, making communication the dominant bottleneck. Moreover, communication can only overlap with the computation of the final micro-batch, leaving the remaining $(n-1)/n$ fraction of computation time underutilized. As a result, even with overlapping techniques, a substantial portion of communication remains non-overlapped.

**Communication Compression.** To further reduce the communication overhead, compression is often applied as an additional step in communication-efficient training. Before transmission,

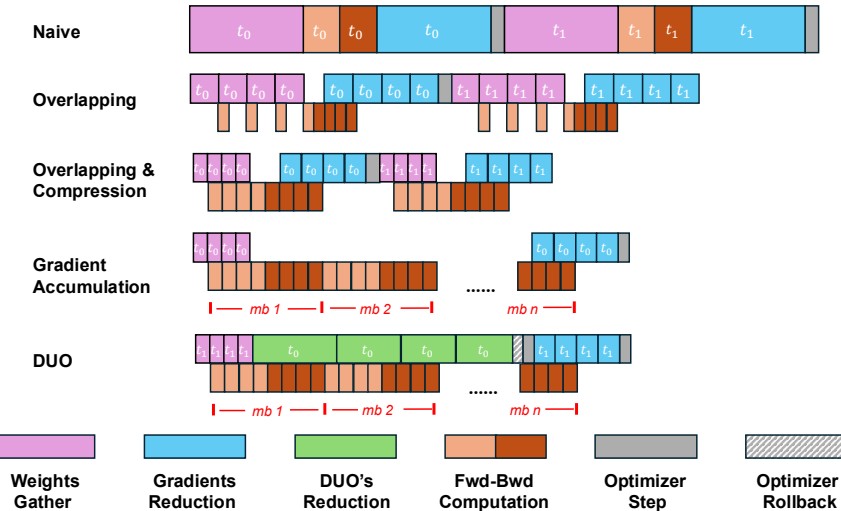

Figure 2: An overview of distributed training with step-by-step integration of communication reduction strategies.

data is first compressed into smaller size, significantly reducing communication volume. Prior works [16, 28, 14] have demonstrated that quantization is an effective method for LLM training, where high-precision weights/gradients are scaled and cast into lower-precision formats. However, achieving a high compression ratio without degrading model accuracy remains a major challenge.

## 2.4 Motivation

By leveraging grouped quantization and a weight-difference communication pattern, communication compression method such as **SDP4Bit** [14] successfully reduces communication to 4-bit precision and is regarded as **a state-of-the-art communication-efficient training framework for Sharded Data Parallelism**. However, it still suffers from accuracy degradation when gradients are compressed to low precision (e.g., 4-bit), despite attempts such as Hadamard transforms to mitigate quantization error. We attribute this performance drop to the high sensitivity of gradients to quantization errors.

Motivated by this limitation, we propose DUO —— a communication-efficient training framework designed to preserve both high accuracy and throughput. DUO achieves accuracy comparable to full-precision baselines, even under ultra-low-bit (1-bit) gradient compression or 0-bit gradient transmission. Additionally, it supports a broader class of compressors such as sparsification, low-rank projection, sign-based compression —— offering greater flexibility for future gradient compression techniques. A formal definition of supported compressors is provided in Definition 4.2.

## 3 The DUO Framework

**DUO** is a system–algorithm co-design framework composed of three components: the base Fast-Slow algorithm (Section 3.1), communication overlapping strategy (Section 3.2), and memory optimization strategy (Section 3.3). Together, these components enable DUO to achieve both high training efficiency and high accuracy. We introduce these three components in the following sections.

### 3.1 Fast-Slow Reduction Algorithm

To improve training accuracy, we leverage the high-precision gradient information that is typically discarded in naive gradient-compression strategies. As highlighted in the red part of Algorithm 1, DUO caches the full-precision gradient $g^p$ before gradient compression (Line 10). This gradient replica will be used for performing high-precision ReduceScatter (Line 5), producing a non-compressed averaged "slow" gradient that compensates for compression error in a later stage (Line 13-15), as explained below.

**Algorithm 1** Distributed training with DUO

**Require:** parameter weight (main copy): $w$, weight for forward-backward: $\tilde{w}$, weight for communication: $w'$, weight difference: $d$, gradient: $g$, worker: $p$, $p$th shard of weight or gradient: $[p]$, gradient produced by worker $p$: $g^p$, gradients with compression $\tilde{g}$, averaged gradient: $\bar{g}$, current iteration: $t$

1: **function** ForwardPass
2:     $d_t[p] = w'_t[p] - \tilde{w}_{t-1}[p]$
3:     $\tilde{d}_t[p] \leftarrow$ QuantizeWeightsDiff$(d_t[p])$
4:     $\tilde{d}_t \leftarrow$ AllGather$(\tilde{d}_t[p])$
5:     Start AsyncReduceScatter$(g^p_{t-1})$
6:     $\tilde{w}_t \leftarrow \tilde{w}_{t-1} + \tilde{d}_t$
7:     $output^p \leftarrow$ ForwardPass$(\tilde{w}_t, input^p)$
8: **function** BackwardPass
9:     $g^p_t \leftarrow$ Gradient$(\tilde{w}_t, output^p_t)$
10:     SaveReplica$(g^p_t)$
11:     $\tilde{g}^p_t \leftarrow$ CompressGradient$(g^p_t)$
12:     $\tilde{\bar{g}}_t[p] \leftarrow$ ReduceScatter$(\tilde{g}^p_t)$
13:     $\bar{g}_{t-1}[p] \leftarrow$ Wait AsyncReduceScatter Finish
14:     Recover $w_{t-1}[p]$
15:     $w_t[p] \leftarrow$ Optimizer$(\bar{g}_{t-1}[p], w_{t-1}[p])$
16:     $w'_{t+1}[p] \leftarrow$ Optimizer$(\tilde{\bar{g}}_t[p], w_t[p])$

---

**Algorithm 2** SGD + Fast-Slow Update

1: Initialize main parameter weights
    $w_0 = \tilde{w}_0$, $g_{t-2} = \mathbf{0}$ for $t = 1$
2: **for all** iteration $t \in [T]$ **do**
3:     Update main weights with full-precision gradient from previous iteration (*slow update*):
    $w_t \leftarrow w_{t-1} - \eta g_{t-2}$
4:     Compute gradient: $g_{t-1} = \nabla f(\tilde{w}_{t-1}; \zeta_{t-1})$
5:     Compress gradient: $\tilde{g}_{t-1} = \mathcal{C}_g(g_{t-1})$
6:     Update temporary weights (*fast update*):
    $w'_t \leftarrow w_t - \eta \tilde{g}_{t-1}$
7:     Compress weight difference:
    $\tilde{\Delta}_t = \mathcal{C}_w(w'_t - \tilde{w}_{t-1})$
8:     Update weights for forward-backward:
    $\tilde{w}_t \leftarrow \tilde{w}_{t-1} + \tilde{\Delta}_t$
9: **end for**

---

To correct the compression error in the "fast" gradients, DUO introduces an optimizer rollback and reset mechanism (Line 14). At each iteration, DUO rolls back the optimizer to restore the main parameters and optimizer states to the state where the compressed gradients are not applied. After that, an additional optimizer step is conducted with the non-compressed averaged gradients (Line 15). By doing so, the optimizer update calculated based on compressed gradients in the last iteration is dropped from the main parameters and the optimizer states, and replaced by the optimizer update based on the non-compressed gradients, resulting in more accurate parameters and states.

The remaining steps of Algorithm 1 (non-red part) follow those of SDP4Bit. The difference between the latest main parameters and the previous model parameters of the local shards on each worker are calculated and compressed (Line 2-3), which is then gathered (Line 4) and used for updating the unsharded model parameters (Line 6) for forward-backward computation (Line 7 and 9). The "fast" gradients are compressed, synchronized, and used for producing a temporary version of the main parameters on the local shard (Line 11, 12, and 16).

## 3.2 Overlapping ReduceScatter with Computation

Although Fast-Slow Reduction improves model training accuracy by incorporating high-precision gradients, it also introduces an additional *ReduceScatter* communication step. Fortunately, this *ReduceScatter* operation can be executed asynchronously alongside the computation.

As illustrated in Algorithm 1, the "slow" gradients $g^p_t$ are cached after they are generated during backpropagation. These full-precision gradients are then averaged via a *ReduceScatter* and used in an additional optimizer step in the next iteration (Line 15). Because the averaged "slow" gradients are not immediately required, the *ReduceScatter* can be executed asynchronously and overlapped with the computation during the interval between SaveReplica (Line 10 of iteration $t$) and optimizer step (Line 15 of iteration $t+1$).

Because the original *AllGather* and *ReduceScatter* operations must complete before the next iteration begins, they take precedence in scheduling. To avoid interference, DUO defers the execution of the high-precision *ReduceScatter* until after the *AllGather* completes (Line 5).

As discussed in Section 2.3, existing overlapping strategies only utilize the computation of the final micro-batch, leaving the rest of the computation time non-overlapped. Thanks to the Fast-

Slow algorithm, the involved *ReduceScatter* can overlap with the computation of all micro-batches, providing significantly greater opportunities for communication overlap.

## 3.3 Memory Footprint Optimization

Implementing the Fast–Slow Reduction algorithm requires a high-precision gradient replica for reduction. This replica is continuously used during the reduction phase. Moreover, during high-precision gradient reduction, the GPU continues performing computations. To integrate Fast-Slow Reduction efficiently without interfering with computation or memory usage, DUO employs a gradient-offloading strategy. Gradients are first copied from device to host memory, where the high-precision reduction is later performed. The averaged gradients are then copied back to the device before executing the optimizer step.

Since CPU memory typically has low bandwidth, a naive device-to-host (D2H) copy before gradient compression would be inefficient, leading to long computation idle times. To achieve near-zero-overhead offloading, DUO employs three complementary techniques:

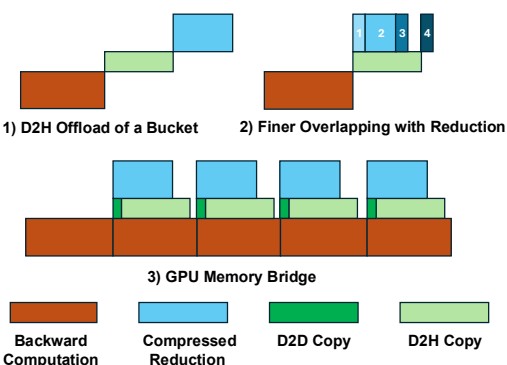

Figure 3: Step-by-step optimization strategy for device to host copy of gradients. First is a naive D2H copy after computation, after adapting with finer reduction and GPU bridge, the D2H can be fully overlapped.

**Bucket-Wise Offloading.** As illustrated in Section 2.3, modern LLM training commonly employs an overlapping strategy to reduce communication overhead. The core idea is to partition computation tasks into *buckets*, where each bucket contains a set of parameters. Gradient reduction begins as soon as the computation for a bucket is completed. Building on this overlapping strategy, we align our gradient offloading granularity with the computation task, introducing *Bucket-Wise Offloading*. In this approach, the gradients for each bucket are offloaded immediately after the corresponding computation finishes, ensuring an early start of D2H transmission.

To accommodate the bucket-wise offloading strategy, we also reorganized the optimizer step to operate at the bucket level. Additionally, we reuse the global gradient norm obtained from the low-precision optimizer step in the previous iteration to ensure consistency of gradient clipping between the two optimizer steps.

**Finer Overlapping with Compressed Gradient Reduction.** A compressed gradient reduction (*fast reduction*) is performed to execute the first-stage optimizer step. As illustrated in the blue region of the second diagram in Figure 3, a typical gradient reduction consists of four phases: (1) compression, (2) point-to-point (P2P) communication, (3) decompression, and (4) local reduction. We observe that only the final phase overwrites the previous gradient values. Therefore, the device-to-host (D2H) transfer can be overlapped with the subsequent compressed gradient reduction at a finer granularity. Therefore, the device-to-host (D2H) transfer can be overlapped with the first three phases of the next compressed gradient reduction—compression, P2P communication, and decompression—thereby reducing transfer overhead. A synchronization barrier is inserted before the local reduction phase to ensure that the D2H transfer has completed.

**GPU Memory Bridge.** Even though we successfully exploit a larger overlapping space to reduce overhead, we observe that the D2H transfer remains too slow to be fully overlapped without blocking the reduction operation. As a result, this bottleneck affects overall training speed.

To address this issue, DUO introduces a *GPU Memory Bridge*, which temporarily caches gradients in a small GPU memory buffer before transferring them to the host memory on the CPU. This strategy acts as an intermediate buffer, effectively serving as a bridge between the GPU and CPU to mitigate its impact on the subsequent reduction.

As illustrated in Figure 3, GPU Memory Bridge extends the available overlapping space with the next bucket's computation, further improving training efficiency. Notably, we set the memory bridge buffer size equal to the bucket size and restrict its usage to a single buffer at any given time to achieve

minimal additional footprint increase. As shown in Table 3, because the bucket sizes are small, this memory increase is negligible.

## 4 Theoretical Analysis

In this section, we establish the theoretical guarantees of convergence for DUO. To simplify the results, we prove the convergence of Algorithm 2, which is equivalent to Algorithm 1 with stochastic gradient descent (SGD) as the optimizer, and hiding the details of sharded DP.

### 4.1 Recursive Error Correction

To make it easier to understand the intuition of the Fast-Slow Reduction algorithm, we present the following equations to show how the compression errors from the weight difference compression and gradient compression are recursively corrected in Algorithm 2.

**Theorem 4.1** (Recursive Error Correction). *The uncompressed weight difference can be recursively expanded as:* $\Delta_t = w_t' - \tilde{w}_{t-1} = -\eta\tilde{g}_{t-1} - (\eta g_{t-2} - \eta\tilde{g}_{t-2}) + (\Delta_{t-1} - \tilde{\Delta}_{t-1})$, *where* $(g_{t-2} - \tilde{g}_{t-2})$ *corrects the error caused by gradient compression, and* $(\Delta_{t-1}) - \tilde{\Delta}_{t-1})$ *corrects the error caused by weight difference compression.*

Theorem 4.1 provides an important intuition for the convergence proof of Algorithm 2. It is indicated that the error of fast-slow update on the parameter weights is composed of two parts: the recursively corrected error of gradient compression, and the recursively corrected error of the weight difference. With the help of this recursive error correction, we can unroll the compression error from the last step back to the first step, and establish the corresponding upper bound.

### 4.2 Convergence Analysis

In our algorithm, we use SGD with gradient and parameter compression to solve the following optimization problem: $f^* = \min_w f(w)$, where $f(w)$ is the objective function, $w \in \mathbb{R}^d$ is the model parameter.

Note that we use arbitrary (potentially biased) compressors for both gradient reduction and parameter weight collection, while SDP4Bit [14] only supports unbiased compressors for the gradients. We formally define the compressors as follows.

**Definition 4.2.** [$\delta$-approximate compressor [15]] An operator $\mathcal{C} : \mathbb{R}^d \to \mathbb{R}^d$ is a $\delta$-approximate compressor for $\delta \in (0, 1]$ if $\|\mathcal{C}(v) - v\|^2 \leq (1 - \delta)\|v\|^2, \forall v \in \mathbb{R}^d$.

We establish our analysis under the following assumptions.

**Assumption 4.3.** For distributed training with $n$ workers, we define the compressed gradient as $\tilde{g}_t = \mathcal{C}_g(g_t) = \frac{1}{n}\sum_{p\in[n]}\mathcal{C}_g'(g_{t,p})$, where $g_t = \frac{1}{n}\sum_{p\in[n]}g_{t,p}$, and $g_{t,p}$ is the stochastic gradient from the $p$th worker in $t$ iteration. We assume that $\mathcal{C}_g$ is a $\delta_g$-approximate compressor of the average gradient $g_t$, and $\mathcal{C}_w$ is a $\delta_w$-approximate compressor for weight difference compression.

**Assumption 4.4.** (Smoothness) We assume that $f(x)$ is $L$-smooth: $\|\nabla f(x) - \nabla f(y)\| \leq L\|x - y\|, \forall x, y \in \mathbb{R}^d$, which implies $f(y) - f(x) \leq \langle \nabla f(x), y - x \rangle + \frac{L}{2}\|y - x\|^2$.

**Assumption 4.5.** For any stochastic gradient $\nabla f(w; \zeta)$, where $\zeta$ is an independent random sample, we assume unbiasedness $\mathbb{E}[\nabla f(w; \zeta)|w] = \nabla f(w)$, and bounded variance $\mathbb{E}[\nabla f(w; \zeta) - \nabla f(w)\|^2|w] \leq \rho\|\nabla f(w)\|^2 + \sigma^2$ ([23], Assumption 3).

We derive the following error bounds on the convergence of Algorithm 2 as follows. All proofs can be found in Appendix A.

**Theorem 4.6** (Convergence error bound). *For arbitrary non-convex function under Assumption 4.3, 4.4, and 4.5, with the $\delta_g$-approximate compressor $\mathcal{C}_g$ for gradient compression and the $\delta_w$-approximate compressor $\mathcal{C}_w$ for weight difference compression, taking learning rate $\eta \leq \sqrt{\frac{\delta_w}{96\alpha L^2(\rho+1)^2}}$, Algorithm 2 converges to a critical point with the following error bound:*

$$\frac{1}{T}\sum_{t=1}^{T}\mathbb{E}\left[\|\nabla f(w_t)\|^2\right] \leq \frac{(f(w_0) - f^*)\sqrt{6144\alpha L^2(\rho+1)^2}}{T\sqrt{\delta_w}}$$
$$+ \frac{\sqrt{384L\sigma^2(4\alpha\eta L + \delta_w)(f(w_0) - f^*)}}{\sqrt{T\delta_w}},$$

*where* $\alpha = \frac{10(1-\delta_g)+8}{\delta_w}$.

Theorem 4.6 shows that our proposed algorithm has the same $\mathcal{O}\left(\frac{1}{\sqrt{T}}\right)$ convergence rate as ordinary SGD for general non-convex functions.

# 5 Experiments

Our experiments are organized into two parts: accuracy and performance. Together, these evaluations show that DUO improves training accuracy while maintaining nearly identical training speed in Sharded Data Parallelism with gradient compression—and can even increase throughput by enabling more aggressive compression.

## 5.1 Environment Setup

**Model and Dataset.** To validate the accuracy improvements of the proposed DUO method, we conduct pre-training experiments on GPT models with sizes ranging from 125M to 6.7B. The hyperparameter settings follow those of the OPT model [35], ensuring that models of the same size use identical configurations. For reproducibility, we provide the detailed configurations in Table 6 (Appendix B).

We use The Pile [12] as our training dataset due to its open-source availability and general applicability. All pre-training tasks are run for 80 000 iterations, processing 80 billion tokens in total. Accuracy is evaluated via validation-loss comparison across all settings.

To effectively evaluate the proposed method, we integrate DUO into Megatron-LM, one of the most commonly used open-source LLM training frameworks.

**Baseline and notation.** We use SDP4Bit—the state-of-the-art low-bit compression strategy combined with Sharded Data Parallelism (ShardedDP) for LLM pre-training—as our baseline. SDP4Bit applies 4-bit randomized symmetric linear quantization to both weight differences and gradients. In our experiments, we retain the same 4-bit weight compression mechanism as in SDP4Bit but vary the gradient quantization across different bit widths. DUO uses 4-bit gradient compression by default. Suffixes such as `G1` denote gradient bit-widths (e.g., DUO-G1 for 1-bit gradients).

**Hardware environment.** The experiments are conducted on three environments:

4-node setup with A100 GPUs: Each node is equipped with 4 NVIDIA A100-SXM4-40GB GPUs connected via NVLink. Nodes are connected by 100 Gbps Slingshot links.

8-node setup with H20 GPUs: Each node is equipped with 8 NVIDIA H20 GPUs connected via NVLink. Nodes are connected by 400 Gbps InfiniBand links.

4-node setup with single A100 GPU: Each node has a single NVIDIA A100-SXM4-40GB GPU. Nodes are connected by 100 Gbps Ethernet links.

Table 2: Final validation loss ↓ for various compression settings.

| Grad Bits | Strategy | 125M | 350M | 1.3B | 6.7B |
|---|---|---|---|---|---|
| 32 | Full Precision | 2.2716 | 2.0582 | 1.8854 | 1.7527 |
| 4 | SDP4Bit | 2.2757 | 2.0629 | 1.8944 | 1.7570 |
| | **DUO** | 2.2727 | 2.0592 | 1.8907 | 1.7535 |
| 1 | SDP4Bit | 2.4204 | 2.1843 | 2.0489 | 1.8434 |
| | **DUO** | 2.2712 | 2.0686 | 1.8943 | 1.7572 |
| 0 | **DUO** | 2.2761 | 2.0628 | 1.8941 | 1.7550 |

## 5.2 E2E Accuracy Evaluation

We evaluate the accuracy of DUO with pre-training tasks. To demonstrate that DUO can achieve high end-to-end training accuracy, we train GPT models of various sizes from scratch.

As shown in Table 2, we assess DUO's performance under different gradient compression ratios. The results indicate a noticeable accuracy gap between SDP4Bit and full-precision training. This

gap becomes more significant as using a more aggressive way for gradient compression or model being larger. In contrast, DUO preserves accuracy even for larger models and ultra low gradient communication bits, maintaining a final loss comparable to the full-precision training.

To further assess DUO under high-compression scenarios, we extend the SDP4Bit setting by reducing the gradient bit-width to 1 or even 0. Note that "0-bit" does not mean that DUO completely removes gradient communication. Instead, the low-precision gradient communication of the "fast" update is removed, but DUO still performs a high-precision (slow) ReduceScatter.

As shown in Table 2, these extreme compressions results in a substantial accuracy drop for SDP4Bit compared to full-precision training. In contrast, DUO maintains high end-to-end accuracy, demonstrating robustness under severe communication constraints.

### 5.3 E2E Performance Test

We evaluate DUO in practical training tasks, focusing on its end-to-end (E2E) speed and memory consumption. We use TFLOPS as the primary metric to measure E2E speed.

**High-bandwidth environment.** For fair comparison, both DUO and SDP4Bit use 4-bit gradient quantization; detailed configuration are in Table 7 (Appendix C). As shown in Table 3, despite introducing an additional high-precision reduction step, DUO achieves training throughput comparable to SDP4Bit. We attribute this to DUO's carefully designed overlapping strategy, which effectively hides communication latency behind computation.

DUO's memory footprint closely matches that of SDP4Bit.. This demonstrating the effectiveness of DUO's high-precision gradient-offloading strategy. The slight increase in memory usage is due to the GPU memory bridge used to mitigate device-to-host transfer overhead. Since the bridge buffer size matches the bucket size and is manually configurable, this overhead does not negatively impact overall performance.

Table 3: E2E performance comparison between DUO and SDP4Bit on two hardware environments: 8×8 H20 nodes and 8×4 A100 nodes. Throughput (TFLOPS ± std) and GPU memory peak (MB).

| Model | Method | H20 (8×8 H20 Nodes) | | A100 (8×4 A100 Nodes) | |
|---|---|---|---|---|---|
| | | Throughput (TFLOPS) | Mem (MB) | Throughput (TFLOPS) | Mem (MB) |
| 1.3B | DUO | $70.82 \pm 0.63$ | 15880 | $117.53 \pm 0.49$ | 21270 |
| | SDP4Bit | $71.02 \pm 0.63$ | 15832 | $117.61 \pm 2.35$ | 20934 |
| 2.7B | DUO | $78.71 \pm 0.92$ | 25600 | $125.31 \pm 0.37$ | 35298 |
| | SDP4Bit | $79.15 \pm 0.80$ | 25640 | $126.42 \pm 0.82$ | 34422 |
| 6.7B | DUO | $84.09 \pm 2.04$ | 16162 | $122.67 \pm 3.76$ | 26480 |
| | SDP4Bit | $85.61 \pm 1.57$ | 15692 | $121.05 \pm 1.89$ | 25684 |
| 13B | DUO | $93.02 \pm 1.08$ | 27596 | $132.88 \pm 0.95$ | 27332 |
| | SDP4Bit | $94.68 \pm 0.77$ | 27220 | $136.73 \pm 1.35$ | 27142 |
| 18B | DUO | $100.85 \pm 1.29$ | 36540 | $116.37 \pm 1.28$ | 27378 |
| | SDP4Bit | $102.93 \pm 1.04$ | 35478 | $122.91 \pm 1.16$ | 27088 |

**Low-bandwidth environment.** We also evaluate DUO in a low-bandwidth cloud environment under various network settings (10 Gbps and 5 Gbps) and gradient compression levels (from 4 bits to 0). As shown in Table 4, DUO consistently achieves end-to-end throughput comparable to SDP4Bit and outperforms the uncompressed baseline. Moreover, DUO-G1 and DUO-G0 achieve higher throughput than SDP4Bit, thanks to reduced gradient communication overhead.

Table 4: End-to-end throughput (TFLOPS) of different methods under constrained network bandwidths using the GPT-350M model with a gradient accumulation step of 32.

| Bandwidth | **DUO-G4** | **DUO-G1** | **DUO-G0** | SDP4Bit | Baseline |
|---|---|---|---|---|---|
| 10 Gbps | $130.81 \pm 3.56$ | $132.67 \pm 2.94$ | $133.10 \pm 4.29$ | $130.52 \pm 2.50$ | $108.89 \pm 2.25$ |
| 5 Gbps | $129.35 \pm 2.96$ | $131.25 \pm 3.16$ | $131.97 \pm 3.06$ | $129.84 \pm 2.97$ | $98.26 \pm 1.32$ |

## 5.4 Ablation Study

**Overlapping capability of DUO's reduction.** DUO strategically overlaps it's newly involved communication with subsequent computations. To hide communication overhead, each iteration's computation time must exceed DUO's high-precision gradient communication time.

The computation-to-communication ratio depends on **network bandwidth, gradient-accumulation steps and sequence length**. As detailed in Section 2.2 and further elaborated in Section 3.2, increasing gradient accumulation steps significantly enhances computation time per iteration:

$$ComputationTimePerIter = AccumulationSteps \times MicrobatchComputationTime \qquad (5)$$

Longer sequence lengths likewise increase computation intensity. Modern LLMs typically use long sequences lengths (e.g., LLaMA 3: 128K, Qwen2: 32K), which substantially increase computation intensity.

We evaluate various combinations of bandwidth, sequence length, and accumulation steps to further demonstrate DUO's overlap effectiveness. As shown in Table 5, both larger accumulation steps and longer sequence lengths contribute to a higher computation-to-communication ratio, resulting in better overlap even under low bandwidth conditions.

Table 5: Computation vs. Communication Overlapping Performance. Breakdown results across bandwidths (10/5/2 Gbps), sequence lengths (2048/4096), and accumulation steps (8/16/32).

| Bandwidth | Seq. Len | Acc. Step | Method | E2E Throughput (TFLOPs) | Grad Comm / Iter (ms) | Comp / Iter (ms) | Full Overlap |
|---|---|---|---|---|---|---|---|
| 10 | 4096 | 32 | DUO | 122.6 | 2135 | 6042 | ✓ |
| | | 32 | SDP4Bit | 123.0 | — | 6042 | |
| | | 16 | DUO | 122.6 | 1959 | 3030 | ✓ |
| | | 16 | SDP4Bit | 123.1 | — | 3030 | |
| | | 8 | DUO | 86.7 | 2029 | 1533 | ✗ |
| | | 8 | SDP4Bit | 117.5 | — | 1533 | |
| | 2048 | 32 | DUO | 83.5 | 2215 | 3024 | ✓ |
| | | 32 | SDP4Bit | 84.4 | — | 3024 | |
| | | 16 | DUO | 71.2 | 2125 | 1545 | ✗ |
| | | 16 | SDP4Bit | 78.8 | — | 1545 | |
| | | 8 | DUO | 39.0 | 2136 | 771 | ✗ |
| | | 8 | SDP4Bit | 71.1 | — | 771 | |
| 5 | 4096 | 32 | DUO | 128.9 | 3033 | 6041 | ✓ |
| | | 32 | SDP4Bit | 129.1 | — | 6041 | |
| | | 16 | DUO | 119.6 | 3137 | 3040 | ✗ |
| | | 16 | SDP4Bit | 123.4 | — | 3040 | |
| | | 8 | DUO | 59.5 | 3287 | 1564 | ✗ |
| | | 8 | SDP4Bit | 112.7 | — | 1564 | |
| | 2048 | 32 | DUO | 80.54 | 2964 | 3012 | ✓ |
| | | 32 | SDP4Bit | 83.4 | — | 3012 | |
| | | 16 | DUO | 50.2 | 3364 | 1520 | ✗ |
| | | 16 | SDP4Bit | 77.3 | — | 1520 | |
| | | 8 | DUO | 22.7 | 3378 | 791 | ✗ |
| | | 8 | SDP4Bit | 65.4 | — | 791 | |
| 2 | 4096 | 32 | DUO | 101.6 | 7973 | 6038 | ✗ |
| | | 32 | SDP4Bit | 119.4 | — | 6038 | |
| | | 16 | DUO | 49.8 | 8198 | 3045 | ✗ |
| | | 16 | SDP4Bit | 105.0 | — | 3045 | |
| | | 8 | DUO | 24.7 | 8273 | 1558 | ✗ |
| | | 8 | SDP4Bit | 84.8 | — | 1558 | |
| | 2048 | 32 | DUO | 41.1 | 7967 | 3048 | ✗ |
| | | 32 | SDP4Bit | 72.7 | — | 3048 | |
| | | 16 | DUO | 20.1 | 8179 | 1566 | ✗ |
| | | 16 | SDP4Bit | 59.9 | — | 1566 | |
| | | 8 | DUO | 10.0 | 8126 | 839 | ✗ |
| | | 8 | SDP4Bit | 43.8 | — | 839 | |

# 6  Conclusion

In this paper, we introduce DUO, a high-precision, communication-efficient training framework. DUO incorporates Fast-Slow, a gradient compression error reduction algorithm, along with a carefully designed optimization strategy. By integrating these components, DUO improves training accuracy to a level comparable to uncompressed training, even at an extreme compression ratio of 0 or 1 bit.

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

## A Proofs

We use the following lemma (simplified from [23], Lemma 14) without proof.

**Lemma A.1.** *For every non-negative sequence $\{r_t\}_{t\geq 0}$ and any parameters $a \geq 0$, $c \geq 0$, $T \geq 0$, there exists a constant $\eta \leq \frac{1}{a}$, such that*

$$\frac{1}{T+1}\sum_{t=0}^{T}\left(\frac{r_t - r_{t+1}}{\eta} + c\eta\right) = \frac{1}{T+1}\frac{r_0 - r_{T+1}}{\eta} + c\eta \leq \frac{ar_0}{T+1} + \frac{2\sqrt{cr_0}}{\sqrt{T+1}}.$$

**Theorem 4.1** (Recursive Error Correction). *The uncompressed weight difference can be recursively expanded as: $\Delta_t = w'_t - \tilde{w}_{t-1} = -\eta\tilde{g}_{t-1} - (\eta g_{t-2} - \eta\tilde{g}_{t-2}) + (\Delta_{t-1} - \tilde{\Delta}_{t-1})$, where $(g_{t-2} - \tilde{g}_{t-2})$ corrects the error caused by gradient compression, and $(\Delta_{t-1}) - \tilde{\Delta}_{t-1})$ corrects the error caused by weight difference compression.*

*Proof.* For the uncompressed weight difference $\Delta_t = w'_t - \tilde{w}_{t-1}$, we can expand it as follows.

$$\begin{aligned}
\Delta_t \\
&= w'_t - \tilde{w}_{t-1} \\
&= w_t - \eta\tilde{g}_{t-1} - \tilde{w}_{t-1} \\
&= w_t - \tilde{w}_{t-1} - \eta\tilde{g}_{t-1}.
\end{aligned}$$

We further expand the term $w_t - \tilde{w}_{t-1}$ as follows.

$$\begin{aligned}
w_t - \tilde{w}_{t-1} \\
&= w_{t-1} - \eta g_{t-2} - \left(\tilde{w}_{t-2} + \tilde{\Delta}_{t-1}\right) \\
&= w_{t-1} - \eta g_{t-2} - \left(\tilde{w}_{t-2} + \Delta_{t-1} + (\tilde{\Delta}_{t-1} - \Delta_{t-1})\right) \\
&= w_{t-1} - \eta g_{t-2} - \left(w'_{t-1} + (\tilde{\Delta}_{t-1} - \Delta_{t-1})\right) \\
&= w_{t-1} - \eta g_{t-2} - \left(w_{t-1} - \eta\tilde{g}_{t-2} + (\tilde{\Delta}_{t-1} - \Delta_{t-1})\right).
\end{aligned}$$

Combining the above equations together, we have

$$\begin{aligned}
\Delta_t \\
&= w_t - \tilde{w}_{t-1} - \eta\tilde{g}_{t-1} \\
&= w_{t-1} - \eta g_{t-2} - \left(w_{t-1} - \eta\tilde{g}_{t-2} + (\tilde{\Delta}_{t-1} - \Delta_{t-1})\right) - \eta\tilde{g}_{t-1} \\
&= -\eta g_{t-2} - \left(-\eta\tilde{g}_{t-2} + (\tilde{\Delta}_{t-1} - \Delta_{t-1})\right) - \eta\tilde{g}_{t-1} \\
&= -\eta\tilde{g}_{t-1} - (\eta g_{t-2} - \eta\tilde{g}_{t-2}) + (\Delta_{t-1} - \tilde{\Delta}_{t-1}).
\end{aligned}$$

$\square$

**Theorem 4.6** (Convergence error bound). *For arbitrary non-convex function under Assumption 4.3, 4.4, and 4.5, with the $\delta_g$-approximate compressor $\mathcal{C}_g$ for gradient compression and the $\delta_w$-approximate compressor $\mathcal{C}_w$ for weight difference compression, taking learning rate $\eta \leq \sqrt{\frac{\delta_w}{96\alpha L^2(\rho+1)^2}}$, Algorithm 2 converges to a critical point with the following error bound:*

$$\begin{aligned}
\frac{1}{T}\sum_{t=1}^{T}\mathbb{E}\left[\|\nabla f(w_t)\|^2\right] &\leq \frac{(f(w_0) - f^*)\sqrt{6144\alpha L^2(\rho+1)^2}}{T\sqrt{\delta_w}} \\
&+ \frac{\sqrt{384L\sigma^2\left(4\alpha\eta L + \delta_w\right)\left(f(w_0) - f^*\right)}}{\sqrt{T\delta_w}},
\end{aligned}$$

*where $\alpha = \frac{10(1-\delta_g)+8}{\delta_w}$.*

*Proof.* Using smoothness (Assumption 4.4), we have

$$f(w_{t+1}) \le f(w_t) - \eta \langle \nabla f(w_t), g_{t-1} \rangle + \frac{\eta^2 L}{2} \|g_{t-1}\|^2.$$

Conditional on $w_t$, taking expectation on the random sample $\zeta_{t-1}$ (denoted by $\mathbb{E}_\zeta$), we have

$$\mathbb{E}_\zeta[f(w_{t+1})]$$

$$\le f(w_t) - \eta \langle \nabla f(w_t), \nabla f(\tilde{w}_{t-1}) \rangle + \frac{\eta^2 L}{2} \mathbb{E}_\zeta \|g_{t-1}\|^2$$

$$= f(w_t) - \eta \langle \nabla f(w_t), \nabla f(\tilde{w}_{t-1}) \rangle + \frac{\eta^2 L}{2} \mathbb{E}_\zeta \|g_{t-1} - \nabla f(\tilde{w}_{t-1}) + \nabla f(\tilde{w}_{t-1})\|^2$$

$$= f(w_t) - \eta \langle \nabla f(w_t), \nabla f(\tilde{w}_{t-1}) \rangle + \frac{\eta^2 L}{2} \mathbb{E}_\zeta \left[ \|g_{t-1} - \nabla f(\tilde{w}_{t-1})\|^2 + \|\nabla f(\tilde{w}_{t-1})\|^2 \right]$$

$$\le f(w_t) - \eta \langle \nabla f(w_t), \nabla f(\tilde{w}_{t-1}) \rangle + \frac{\eta^2 L(\rho+1)}{2} \|\nabla f(\tilde{w}_{t-1})\|^2 + \frac{\eta^2 L \sigma^2}{2}$$

$$= f(w_t) - \eta \langle \nabla f(w_t), \nabla f(\tilde{w}_{t-1}) - \nabla f(w_t) + \nabla f(w_t) \rangle$$
$$\quad + \frac{\eta^2 L(\rho+1)}{2} \|\nabla f(\tilde{w}_{t-1}) - \nabla f(w_t) + \nabla f(w_t)\|^2 + \frac{\eta^2 L \sigma^2}{2}$$

$$\le f(w_t) - \eta \langle \nabla f(w_t), \nabla f(\tilde{w}_{t-1}) - \nabla f(w_t) \rangle - \eta \|\nabla f(w_t)\|^2$$
$$\quad + \eta^2 L(\rho+1) \|\nabla f(\tilde{w}_{t-1}) - \nabla f(w_t)\|^2 + \eta^2 L(\rho+1) \|\nabla f(w_t)\|^2 + \frac{\eta^2 L \sigma^2}{2}$$

$$= f(w_t) - \eta \left(1 - \eta L(\rho+1)\right) \|\nabla f(w_t)\|^2 + \frac{\eta^2 L \sigma^2}{2}$$
$$\quad - \eta \langle \nabla f(w_t), \nabla f(\tilde{w}_{t-1}) - \nabla f(w_t) \rangle + \eta^2 L(\rho+1) \|\nabla f(\tilde{w}_{t-1}) - \nabla f(w_t)\|^2$$

$$\le f(w_t) - \eta \left(1 - \eta L(\rho+1)\right) \|\nabla f(w_t)\|^2 + \frac{\eta^2 L \sigma^2}{2}$$
$$\quad + \frac{\eta}{2} \|\nabla f(w_t)\|^2 + \frac{\eta}{2} \|\nabla f(\tilde{w}_{t-1}) - \nabla f(w_t)\|^2 \qquad \triangleright \langle a, b \rangle \le \tfrac{1}{2}\|a\|^2 + \tfrac{1}{2}\|b\|^2$$
$$\quad + \eta^2 L(\rho+1) \|\nabla f(\tilde{w}_{t-1}) - \nabla f(w_t)\|^2$$

$$= f(w_t) - \eta \left( \frac{1}{2} - \eta L(\rho+1) \right) \|\nabla f(w_t)\|^2 + \left( \frac{\eta}{2} + \eta^2 L(\rho+1) \right) \|\nabla f(\tilde{w}_{t-1}) - \nabla f(w_t)\|^2$$
$$\quad + \frac{\eta^2 L \sigma^2}{2}.$$

Again using smoothness, and taking $\eta \le \frac{1}{4L(\rho+1)}$ which leads to $-\eta \left( \frac{1}{2} - \eta L(\rho+1) \right) \le -\frac{\eta}{4}$ and $\frac{\eta}{2} + \eta^2 L(\rho+1) \le \frac{3\eta}{4}$, we have

$$\mathbb{E}_\zeta[f(w_{t+1})]$$

$$\le f(w_t) - \frac{\eta}{4} \|\nabla f(w_t)\|^2 + \frac{\eta}{4} \|\nabla f(\tilde{w}_{t-1}) - \nabla f(w_t)\|^2 + \frac{\eta^2 L \sigma^2}{2}$$

$$\le f(w_t) - \frac{\eta}{4} \|\nabla f(w_t)\|^2 + \frac{3\eta L^2}{4} \|w_t - \tilde{w}_{t-1}\|^2 + \frac{\eta^2 L \sigma^2}{2}.$$

Now we establish the upper bound of $\|w_t - \tilde{w}_{t-1}\|^2$ as follows.

First, using $w_t = w_{t-1} - \eta g_{t-2}$ and $\tilde{w}_{t-1} = \tilde{w}_{t-2} + \tilde{\Delta}_{t-1}$, we have the following equations:

$$w_t - \tilde{w}_{t-1}$$

$$= w_{t-1} - \eta g_{t-2} - \tilde{w}_{t-2} - \tilde{\Delta}_{t-1}$$

$$= w_{t-1} - \eta g_{t-2} - \tilde{w}_{t-2} - \mathcal{C}_w(w'_{t-1} - \tilde{w}_{t-2})$$

$$= w_{t-1} - \eta g_{t-2} - \tilde{w}_{t-2} - \mathcal{C}_w(w_{t-1} - \eta \tilde{g}_{t-2} - \tilde{w}_{t-2})$$

$$= w_{t-1} - \eta \tilde{g}_{t-2} - \tilde{w}_{t-2} - \mathcal{C}_w(w_{t-1} - \eta \tilde{g}_{t-2} - \tilde{w}_{t-2}) + \eta \tilde{g}_{t-2} - \eta g_{t-2}.$$

For $\forall a, b > 0$, we have

$$
\begin{aligned}
&\|w_t - \tilde{w}_{t-1}\|^2 \\
&= \|w_{t-1} - \eta\tilde{g}_{t-2} - \tilde{w}_{t-2} - \mathcal{C}_w(w_{t-1} - \eta\tilde{g}_{t-2} - \tilde{w}_{t-2}) + \eta\tilde{g}_{t-2} - \eta g_{t-2}\|^2 \\
&\leq [\|w_{t-1} - \eta\tilde{g}_{t-2} - \tilde{w}_{t-2} - \mathcal{C}_w(w_{t-1} - \eta\tilde{g}_{t-2} - \tilde{w}_{t-2})\| + \|\eta\tilde{g}_{t-2} - \eta g_{t-2}\|]^2 \\
&\leq (1+a)\|w_{t-1} - \eta\tilde{g}_{t-2} - \tilde{w}_{t-2} - \mathcal{C}_w(w_{t-1} - \eta\tilde{g}_{t-2} - \tilde{w}_{t-2})\|^2 \\
&\quad + (1+1/a)\|\eta\tilde{g}_{t-2} - \eta g_{t-2}\|^2 \qquad\qquad \triangleright xy = \sqrt{a}x\frac{y}{\sqrt{a}} \leq \frac{ax^2}{2} + \frac{y^2}{2a}, \forall x, y \\
&\leq (1+a)(1-\delta_w)\|w_{t-1} - \eta\tilde{g}_{t-2} - \tilde{w}_{t-2}\|^2 + (1+1/a)\|\eta\tilde{g}_{t-2} - \eta g_{t-2}\|^2 \\
&\leq (1+a)(1-\delta_w)(1+b)\|w_{t-1} - \tilde{w}_{t-2}\|^2 \\
&\quad + (1+a)(1-\delta_w)(1+1/b)\|\eta\tilde{g}_{t-2}\|^2 + (1+1/a)\|\eta\tilde{g}_{t-2} - \eta g_{t-2}\|^2 \\
&\leq (1+a)(1-\delta_w)(1+b)\|w_{t-1} - \tilde{w}_{t-2}\|^2 \\
&\quad + 2(1+a)(1-\delta_w)(1+1/b)\eta^2\|g_{t-2}\|^2 + 2(1+a)(1-\delta_w)(1+1/b)\eta^2\|\tilde{g}_{t-2} - g_{t-2}\|^2 \\
&\quad + (1+1/a)\eta^2\|\tilde{g}_{t-2} - g_{t-2}\|^2.
\end{aligned}
$$

Also, taking $\delta$-approximation w.r.t. the gradient compressor $\mathcal{C}_g$, we have

$$
\begin{aligned}
&\|w_t - \tilde{w}_{t-1}\|^2 \\
&\leq (1+a)(1-\delta_w)(1+b)\|w_{t-1} - \tilde{w}_{t-2}\|^2 \\
&\quad + 2(1+a)(1-\delta_w)(1+1/b)\eta^2\|g_{t-2}\|^2 \\
&\quad + [(1+1/a) + 2(1+a)(1-\delta_w)(1+1/b)]\,\eta^2\|\tilde{g}_{t-2} - g_{t-2}\|^2 \\
&\leq (1+a)(1-\delta_w)(1+b)\|w_{t-1} - \tilde{w}_{t-2}\|^2 \\
&\quad + 2(1+a)(1-\delta_w)(1+1/b)\eta^2\|g_{t-2}\|^2 \\
&\quad + [(1+1/a) + 2(1+a)(1-\delta_w)(1+1/b)]\,\eta^2(1-\delta_g)\|g_{t-2}\|^2 \\
&= (1+a)(1-\delta_w)(1+b)\|w_{t-1} - \tilde{w}_{t-2}\|^2 \\
&\quad + [2(1+a)(1-\delta_w)(1+1/b) + (1+1/a)(1-\delta_g)]\,\eta^2\|g_{t-2}\|^2 \\
&\quad + [2(1+a)(1-\delta_w)(1+1/b)(1-\delta_g)]\,\eta^2\|g_{t-2}\|^2
\end{aligned}
$$

Further, taking expectation w.r.t. the random sample $\zeta_{t-2}$ (denoted by $\mathbb{E}_\zeta$), and denoting $\alpha = 2(1+a)(1-\delta_w)(1+1/b) + (1+1/a)(1-\delta_g) + 2(1+a)(1-\delta_w)(1+1/b)(1-\delta_g)$, we have

$$
\begin{aligned}
&\mathbb{E}_\zeta\left[\|w_t - \tilde{w}_{t-1}\|^2\right] \\
&\leq (1+a)(1-\delta_w)(1+b)\|w_{t-1} - \tilde{w}_{t-2}\|^2 + \alpha\eta^2\left[\|\nabla f(\tilde{w}_{t-2})\|^2 + \mathbb{E}_\zeta\left[\|g_{t-2} - \nabla f(\tilde{w}_{t-2})\|^2\right]\right] \\
&\leq (1+a)(1-\delta_w)(1+b)\|w_{t-1} - \tilde{w}_{t-2}\|^2 + \alpha\eta^2(\rho+1)\|\nabla f(\tilde{w}_{t-2})\|^2 + \alpha\eta^2\sigma^2.
\end{aligned}
$$

Then, taking $a = \frac{\delta_w}{2(1-\delta_w)}$, $b = \frac{\delta_w}{2(2-\delta_w)}$, we have

$$
\begin{aligned}
&\mathbb{E}_\zeta\left[\|w_t - \tilde{w}_{t-1}\|^2\right] \\
&\leq (1-\delta_w/4)\|w_{t-1} - \tilde{w}_{t-2}\|^2 + \alpha\eta^2(\rho+1)\|\nabla f(\tilde{w}_{t-2})\|^2 + \alpha\eta^2\sigma^2 \\
&\leq (1-\delta_w/4)\|w_{t-1} - \tilde{w}_{t-2}\|^2 + 2\alpha\eta^2(\rho+1)\|\nabla f(w_{t-1})\|^2 \\
&\quad + 2\alpha\eta^2(\rho+1)\|\nabla f(\tilde{w}_{t-2}) - \nabla f(w_{t-1})\|^2 + \alpha\eta^2\sigma^2 \\
&\leq (1-\delta_w/4)\|w_{t-1} - \tilde{w}_{t-2}\|^2 + 2\alpha\eta^2(\rho+1)\|\nabla f(w_{t-1})\|^2 + 2\alpha\eta^2(\rho+1)L^2\|w_{t-1} - \tilde{w}_{t-2}\|^2 \\
&\quad + \alpha\eta^2\sigma^2 \qquad\qquad\qquad\qquad\qquad\qquad\qquad\qquad\qquad\qquad\qquad \triangleright \text{smoothness} \\
&\leq (1-\delta_w/8)\|w_{t-1} - \tilde{w}_{t-2}\|^2 + 2\alpha\eta^2(\rho+1)\|\nabla f(w_{t-1})\|^2 + \alpha\eta^2\sigma^2, \\
&\qquad\qquad\qquad\qquad\qquad \triangleright \text{using } \eta \leq \sqrt{\frac{\delta_w}{16\alpha(\rho+1)L^2}}, \text{ i.e. } 2\alpha\eta^2(\rho+1)L^2 \leq \frac{\delta_w}{8}
\end{aligned}
$$

where $\alpha$ could be simplified as

$$\alpha$$

$$= 2(1+a)(1-\delta_w)(1+1/b) + (1+1/a)(1-\delta_g) + 2(1+a)(1-\delta_w)(1+1/b)(1-\delta_g)$$

$$= \frac{(2-\delta_w)(4-\delta_w)}{\delta_w} + \frac{(2-\delta_w)(1-\delta_g)}{\delta_w} + \frac{(2-\delta_w)(4-\delta_w)(1-\delta_g)}{\delta_w}$$

$$\leq \frac{10(1-\delta_g)+8}{\delta_w}.$$

We simplify the notation by denoting $\mathbb{E}[\cdot] = \mathbb{E}_\zeta[\cdot]$, and unroll the sequence back to $t = 2$, which yields:

$$\mathbb{E}\left[\|w_t - \tilde{w}_{t-1}\|^2\right]$$

$$\leq \sum_{\tau=2}^{t}(1-\delta_w/8)^{t-\tau}\left[2\alpha\eta^2(\rho+1)\|\nabla f(w_{\tau-1})\|^2 + \alpha\eta^2\sigma^2\right] \quad \triangleright w_1 - \tilde{w}_0 = w_1 - w_0 = \mathbf{0}$$

$$\leq 2\alpha\eta^2(\rho+1)\sum_{\tau=2}^{t}(1-\delta_w/8)^{t-\tau}\left[\|\nabla f(w_{\tau-1})\|^2\right] + \alpha\eta^2\sigma^2\sum_{\tau=0}^{t}(1-\delta_w/8)^{t-\tau}$$

$$\leq 2\alpha\eta^2(\rho+1)\sum_{\tau=2}^{t}(1-\delta_w/8)^{t-\tau}\left[\|\nabla f(w_{\tau-1})\|^2\right] + \frac{8\alpha\eta^2\sigma^2}{\delta_w}.$$

$$\triangleright \sum_{\tau=0}^{t}(1-\frac{\delta_w}{8})^{t-\tau} \leq \frac{1}{1-(1-\frac{\delta_w}{8})}$$

Then, staking $\mathbb{E}\left[\|w_t - \tilde{w}_{t-1}\|^2\right]$ and taking total expectation, we have

$$\sum_{t=1}^{T}\mathbb{E}\left[\|w_t - \tilde{w}_{t-1}\|^2\right]$$

$$\leq 2\alpha\eta^2(\rho+1)\sum_{t=2}^{T}\sum_{\tau=2}^{t}(1-\delta_w/8)^{t-\tau}\left[\|\nabla f(w_{\tau-1})\|^2\right] + \frac{8T\alpha\eta^2\sigma^2}{\delta_w}$$

$$\leq 2\alpha\eta^2(\rho+1)\sum_{t=2}^{T}\left[\sum_{\tau=0}^{+\infty}(1-\delta_w/8)^{\tau}\right]\|\nabla f(w_{t-1})\|^2 + \frac{8T\alpha\eta^2\sigma^2}{\delta_w}$$

$$\leq \frac{16\alpha\eta^2(\rho+1)}{\delta_w}\sum_{t=2}^{T}\|\nabla f(w_{t-1})\|^2 + \frac{8T\alpha\eta^2\sigma^2}{\delta_w}$$

$$= \frac{16\alpha\eta^2(\rho+1)}{\delta_w}\sum_{t=1}^{T-1}\|\nabla f(w_t)\|^2 + \frac{8T\alpha\eta^2\sigma^2}{\delta_w}$$

$$\leq \frac{16\alpha\eta^2(\rho+1)}{\delta_w}\sum_{t=1}^{T}\|\nabla f(w_t)\|^2 + \frac{8T\alpha\eta^2\sigma^2}{\delta_w}$$

Putting all the ingredients together and taking total expectation, we have

$$
\sum_{t=1}^{T} \mathbb{E}[f(w_{t+1})]
$$

$$
\leq \sum_{t=1}^{T} \mathbb{E}[f(w_t)] - \frac{\eta}{4} \sum_{t=1}^{T} \mathbb{E}\left[\|\nabla f(w_t)\|^2\right] + \frac{3\eta L^2}{4} \sum_{t=1}^{T} \mathbb{E}\left[\|w_t - \tilde{w}_{t-1}\|^2\right] + \frac{T\eta^2 L\sigma^2}{2}
$$

$$
\leq \sum_{t=1}^{T} \mathbb{E}[f(w_t)] - \frac{\eta}{4} \sum_{t=1}^{T} \mathbb{E}\left[\|\nabla f(w_t)\|^2\right] + \frac{3\eta L^2}{4} \left[ \frac{16\alpha\eta^2(\rho+1)}{\delta_w} \sum_{t=1}^{T} \|\nabla f(w_t)\|^2 + \frac{8T\alpha\eta^2\sigma^2}{\delta_w} \right]
$$

$$
+ \frac{T\eta^2 L\sigma^2}{2}
$$

$$
\leq \sum_{t=1}^{T} \mathbb{E}[f(w_t)] - \frac{\eta}{8} \sum_{t=1}^{T} \mathbb{E}\left[\|\nabla f(w_t)\|^2\right] + \frac{6T\alpha\eta^3 L^2\sigma^2}{\delta_w} + \frac{T\eta^2 L\sigma^2}{2}.
$$

$$
\triangleright \text{ using } \eta \leq \sqrt{\frac{\delta_w}{96 L^2 \alpha(\rho+1)}}, \text{ i.e. } \frac{3\eta L^2}{4} \frac{16\alpha\eta^2(\rho+1)}{\delta_w} \leq \frac{\eta}{8}
$$

Re-arranging the terms, we have

$$
\frac{\eta}{8} \sum_{t=1}^{T} \mathbb{E}\left[\|\nabla f(w_t)\|^2\right] \leq \mathbb{E}[f(w_0)] - \mathbb{E}[f(w_T)] + \frac{6T\alpha\eta^3 L^2\sigma^2}{\delta_w} + \frac{T\eta^2 L\sigma^2}{2}
$$

$$
\triangleright \text{ note that } w_1 = w_0
$$

$$
\Rightarrow \quad \frac{1}{T} \sum_{t=1}^{T} \mathbb{E}\left[\|\nabla f(w_t)\|^2\right] \leq \frac{8\mathbb{E}[f(w_0) - f(w_T)]}{T\eta} + \frac{48\alpha\eta^2 L^2\sigma^2}{\delta_w} + 4\eta L\sigma^2.
$$

Finally, using Lemma A.1, we have

$$
\frac{1}{T} \sum_{t=1}^{T} \mathbb{E}\left[\|\nabla f(w_t)\|^2\right]
$$

$$
\leq \frac{8\mathbb{E}[f(w_0) - f(w_T)]}{T\eta} + \frac{16\alpha\eta^2 L^2\sigma^2}{\delta_w} + 4\eta L\sigma^2
$$

$$
\leq \frac{(f(w_0) - f^*)\sqrt{6144\alpha L^2(\rho+1)^2}}{T\sqrt{\delta_w}} + \frac{\sqrt{384 L\sigma^2 (4\alpha\eta L + \delta_w)(f(w_0) - f^*)}}{\sqrt{T\delta_w}}.
$$

$$
\triangleright \text{ taking } \eta \leq \sqrt{\frac{\delta_w}{96\alpha L^2(\rho+1)^2}} \leq \min\left( \frac{1}{4L(\rho+1)}, \sqrt{\frac{\delta_w}{16\alpha(\rho+1)L^2}}, \sqrt{\frac{\delta_w}{96 L^2\alpha(\rho+1)}} \right)
$$

$\square$

# B   Training Configuration for the E2E Accuracy Test

Table 6: E2E accuracy task training parameters.

| Model Size | Learning Rate | Betas | Epsilon | Weight Decay | Batch Size |
|---|---|---|---|---|---|
| 125M | 6e-4 | 0.9, 0.95 | 1e-8 | 0.1 | 512 |
| 350M | 3e-4 | 0.9, 0.95 | 1e-8 | 0.1 | 512 |
| 1.3B | 2e-4 | 0.9, 0.95 | 1e-8 | 0.1 | 512 |
| 6.7B | 1.2e-4 | 0.9, 0.95 | 1e-8 | 0.1 | 512 |

# C   Training Configuration for the Throughput Test

Table 7: Training configuration for the throughput test on A100 and H20 systems.

| System | Parameter | 1.3B | 2.7B | 6.7B | 13B | 18B |
|--------|-----------|------|------|------|-----|-----|
| A100 (8×4) | TP Size | 4 | 4 | 4 | 4 | 4 |
| | PP Size | 1 | 1 | 1 | 2 | 2 |
| | DP Size | 8 | 8 | 8 | 4 | 4 |
| | Grad Accum Step | 32 | 32 | 32 | 32 | 32 |
| | Microbatch Size | 4 | 4 | 1 | 1 | 1 |
| | Seq. Length | 4096 | 4096 | 4096 | 4096 | 2048 |
| H20 (8×8) | TP Size | 8 | 8 | 8 | 8 | 8 |
| | PP Size | 1 | 1 | 1 | 1 | 1 |
| | DP Size | 8 | 8 | 8 | 8 | 8 |
| | Grad Accum Step | 32 | 32 | 32 | 32 | 32 |
| | Microbatch Size | 4 | 4 | 1 | 1 | 1 |
| | Seq. Length | 4096 | 4096 | 4096 | 4096 | 4096 |

