# OpenReview forum: "DUO: No Compromise to Accuracy Degradation"
_NeurIPS.cc/2025/Conference — NeurIPS 2025 poster_

### Official Review · Reviewer_o9Va · 2025-06-29

**Clarity:** 3
**Significance:** 4
**Originality:** 3
**Rating:** 6
**Confidence:** 3

**Summary:**

This paper introduces DUO (Double Update Overlap), a distributed training framework designed to mitigate accuracy degradation caused by gradient compression, particularly at low precision (e.g., 4-bit or 1-bit). DUO achieves this by inserting an additional high-precision gradient synchronization step into a computation-only phase, fully hiding its communication overhead. The framework leverages a fast-slow reduction algorithm that caches full precision gradients for a later high-precision update. The main advantage is that such a new algorithm is amenable to communication overlapping.

The authors also incorporate several interesting memory optimization strategies (Bucket-Wise Offloading, Finer Overlapping and GPU Memory Bridge). Theoretical convergence guarantees are provided followed by extensive experimental validation on LLM models, showing that DUO restores accuracy comparable to uncompressed training even under extreme compression (1-bit and 0-bit gradients), while maintaining comparable throughput and memory usage to state-of-the-art methods like SDP4Bit.

**Questions:**

Few quick questions to the authors:
- Could the authors provide more insight into the computational cost of the optimizer rollback and reset mechanism (Line 14 of Algorithm 1) and if it contributes to any minor overhead not captured in the E2E throughput or experiment descriptions?

- The concept of 0-bit transmission seems intriguing to me as well as bit confusing. Can the authors offer more explanation on why DUO performs the way it does in this setting? What brings the robustness?

- Have the authors thought about how DUO can be applied to other important distributed training paradigms such as Federated learning, Semi-Sync Training (e.g. DiLoCo) etc? Curious to hear any thoughts on these.

- Are there any specific types of models or training scenarios where DUO might not be as effective, or where its benefits might be less pronounced? Have the authors envisioned such setups?

**Ethical Concerns:**

["NO or VERY MINOR ethics concerns only"]

**Final Justification:**

== Update after rebuttal ==

Raised from 5 to 6.

**Limitations:**

The paper only provides a novel algorithm, doesn't release any dataset or model. Therefore I think this question does not fully apply here. The authors have also added this justification.

**Paper Formatting Concerns:**

- The terms full-precision and non-compressed are used interchangeably in the paper and it leads to some confusion and mental overheads in absorbing the material. Helpful if the authors stick to one convention consistently (e.g. simply use compressed vs non-compressed throughout?)

- typos and language issues: encourage authors to use a grammar check tool and fix the typos. I found a few in the paper.

e.g.

line 307 -> aggressive misspelled

line 458 -> it's -> its

**Quality:**

3

**Strengths And Weaknesses:**

Strengths:

Overall, I feel this paper is a strong empirical contribution, solving a key challenge in distributed training.
Below I highlight a few of them:
- The paper tackles the key challenge in memory-efficient (low-bit) LLM training these days - which is the significant drop in accuracy. This has been a long-standing (and largely unsolved) bottleneck. Techniques  provided in this paper seem very beneficial to me to tackling these challenges..
- I found the core idea of using a high-precision "slow" gradient update alongside a low-precision "fast" update, and cleverly overlapping the slow gradient communication innovative.
- The experimental results presented are strong. DUO demonstrates impressive accuracy restoration across various GPT model sizes and extreme compression ratios (including 0-bit gradients), almost matching full-precision baselines. This is a very compelling result.
- The authors further optimize their solutions by incorporating more practical systems optimization solutions such: Bucket-Wise Offloading, Finer Overlapping, GPU Memory Bridge - these are great contributions that are well-explained and address practical implementation challenges.
- I like the fact that the authors use Megatron-LM as the reference open-source integration. This makes their approach well tested and can increases their adoption potentially.

Weaknesses:
- While effective, the DUO framework, particularly the Fast-Slow Reduction algorithm with its optimizer rollback and reset mechanism (Line 14-15 of Algorithm 1) appear quite intricate. This complexity might pose a challenge for widespread adoption or implementation outside of highly specialized teams. Wondering if the authors have thought about this? Are there other ways this could be simplified?
- There has been a major pillar of work in compressed communication for distributed learning which focusses on Error Feedback mechanisms. While the paper mentions error feedback (EF) methods and shows SDP4Bit-G1-EF failing to converge, a more in-depth comparison or discussion on why DUO succeeds where EF struggles, especially for LLMs under aggressive compression, could strengthen the argument.
- While extensive experiments on GPT models are valuable since dense models form the bulk of LLMs used these days, a brief discussion on the potential applicability or challenges of DUO to other deep learning architectures (e.g. sparse MoE type models) could be beneficial. Not sure if this has been discussed in the paper and I missed it.

---

> ### Author Rebuttal · Authors · 2025-07-30
>
> **W1 - While effective, the DUO framework, particularly the Fast-Slow Reduction algorithm with its optimizer rollback and reset mechanism (Line 14-15 of Algorithm 1) appear quite intricate. This complexity might pose a challenge for widespread adoption or implementation outside of highly specialized teams. Wondering if the authors have thought about this? Are there other ways this could be simplified?**
>
> The rollback step is **unavoidable** if we want the high‑precision correction to apply to **exactly the same optimizer states used** in the low‑precision update. After the compressed update advances the optimizer state, we must restore that state to its pre‑update values before applying the accurate version.
>
> Fortunately, because **Adam‑style optimizers** update their statistics element‑wise, this rollback is **analytically reversible** and introduces **negligible overhead** (see also Q1). We have also explored implementation simplifications: this function can be implemented using frameworks like *Torch Compiler* or *Triton*, but for ease of adoption, we have implemented a **self‑contained CUDA kernel** that fuses the rollback and accurate re-apply into a single call.
>
> This kernel will be open-sourced upon acceptance, allowing practitioners to integrate it into existing Adam/AdamW pipelines with just a **two-line wrapper**, requiring no major refactoring.
>
> In short, while the rollback procedure is essential to ensure Fast–Slow’s **accuracy guarantee**, we have packaged it for **practical use**, so that non-specialized teams can adopt DUO with **minimal engineering burden**.
>
>
> **W2 - There has been a major pillar of work in compressed communication for distributed learning which focusses on Error Feedback mechanisms. While the paper mentions error feedback (EF) methods and shows SDP4Bit-G1-EF failing to converge, a more in-depth comparison or discussion on why DUO succeeds where EF struggles, especially for LLMs under aggressive compression, could strengthen the argument.**
>
> In our view, the failure of traditional **Error Feedback (EF)** mechanisms under aggressive compression may stem from their incompatibility with the **Adam optimizer** and **large scale Lange Model**. Prior EF work [1] (e.g., using SignSGD) is based on **SGD-style optimizers** on **relatively small CV models**, where EF has shown strong empirical results. However, EF appears to perform poorly when combined with Adam or AdamW for large scale LLM pretraining.
>
> That said, this remains a hypothesis, and we acknowledge that a rigorous theoretical and empirical study is needed to fully understand why EF struggles in this context. Our **DUO framework**, in contrast, is explicitly designed to operate within the **Adam + SDP** setting and includes **system-level integration** to ensure high-precision correction is applied effectively without optimizer-state inconsistency.
>
> We thank the reviewer for highlighting this point, and we agree that a deeper investigation into EF’s limitations under modern LLM training regimes would be valuable future work.
>
> [1] Karimireddy, Sai Praneeth, et al. "Error feedback fixes signsgd and other gradient compression schemes." *International Conference on Machine Learning*. PMLR, 2019.
>
> **W3 - While extensive experiments on GPT models are valuable since dense models form the bulk of LLMs used these days, a brief discussion on the potential applicability or challenges of DUO to other deep learning architectures (e.g. sparse MoE type models) could be beneficial. Not sure if this has been discussed in the paper and I missed it.**
>
> Thank you for raising this point. We did not conduct a comprehensive evaluation on MoE-style models due to limited compute resources, but we agree this is an important direction.
>
> That said, DUO is not restricted to dense, fully deterministic Transformer blocks. The prerequisite is that the training stack employs **Sharded Data Parallelism (SDP)** using ReduceScatter/AllGather for gradient synchronization—this is the default in Megatron-LM and similar frameworks.
>
> Under this communication pattern, DUO is directly applicable to sparse expert architectures, including Mixture-of-Experts (MoE) models and MoE-style variants of LLaMA, Qwen, and other recent large models.
>
> **Q1- Could the authors provide more insight into the computational cost of the optimizer rollback and reset mechanism (Line 14 of Algorithm 1) and if it contributes to any minor overhead not captured in the E2E throughput or experiment descriptions?**
>
>   The computational overhead of the optimizer rollback is very similar to that of a standard optimizer step—both are **lightweight, element-wise operations** with no inter-element dependencies. As such, their cost is **negligible** compared to the overall training step.
>
>   To provide concrete numbers, we profiled the optimizer rollback during GPT-350M training with 8 gradient accumulation steps and a sequence length of 2048. In this setting, the forward-backward computation took approximately **771 ms per iteration**, while the rollback step required only **2.01 ms**.
>
> This overhead is therefore negligible relative to the total end-to-end training time. We will clarify this point in the final version and include profiling results to support our claim.
>
> **Q2 - The concept of 0-bit transmission seems intriguing to me as well as bit confusing. Can the authors offer more explanation on why DUO performs the way it does in this setting? What brings the robustness?**
>
> In the full **DUO** setting, we maintain two complementary gradient communication channels, as shown in Algorithm 1 (Lines 5 and 12):
> 1) **Timely but compressed communication**
> 2) **Accurate but one‑step‑stale asynchronous communication**
>
> When both channels are active (e.g., compressing the first gradient stream to 4 bits), the model receives **fresh gradient information every step** and an **exact correction one step later**. Resulting in final accuracy that nearly matches the full-precision baseline (as shown in Table 2).
>
> However, if the cluster’s network is so bandwidth-constrained that even 4‑bit communication becomes a bottleneck, the compressed communication channel can be disabled—resulting in the **“0-bit” transmission**. In this variant, DUO relies solely on the delayed but accurate gradient updates. This still leads to **acceptable convergence**, although typically with a **slight drop in accuracy** compared to the full DUO setup.
>
> As discussed in **Appendix D**, the optimal balance between compressed and accurate channels depends on factors such as **model size, available bandwidth, sequence length**, and the **specific compressor used**. Because no single setting is universally best, DUO is designed to expose both communication channels, allowing practitioners to **tune the trade-off** between communication budget and final accuracy based on their system constraints.
>
> **Q3 - Have the authors thought about how DUO can be applied to other important distributed training paradigms such as Federated learning, Semi-Sync Training (e.g. DiLoCo) etc? Curious to hear any thoughts on these.**
>
> Yes, **DUO** can be applied in conjunction with other training paradigms such as **Federated Learning** or **LocalSGD-style training** (e.g., **DiLoCo**).
>
> In particular, DiLoCo and similar LocalSGD-based methods are **orthogonal to DUO**, as LocalSGD primarily targets **DDP-style training**, while DUO is designed for **Sharded Data Parallelism (SDP)**. The combination of Data Parallelism (DDP) and Sharded Data Parallelism (SDP) has already been explored in prior work such as **HSDP** [1]. In that setup, combining DUO with DiLoCo enables a two-pronged strategy:
> - DiLoCo reduces **inter-replica communication** by lowering synchronization frequency
> - DUO reduces **intra-replica communication** through error-compensated compression
>
> To elaborate: LocalSGD-style methods aim to reduce communication by allowing several local update steps before synchronization. These typically assume that each worker holds the entire model (or identical copies), which differs fundamentally from SDP, where model parameters and optimizer states are **sharded across workers**, and communication relies on **ReduceScatter/AllGather**.
>
> However, with the right system design—such as that used in HSDP—**LocalSGD and DUO can be combined effectively**, addressing distinct communication bottlenecks without conflict.
>
> [1] HSDP: Accelerating Large-scale Model Training via Efficient Sharded Data Parallelism
>
> **Q4 - Are there any specific types of models or training scenarios where DUO might not be as effective, or where its benefits might be less pronounced? Have the authors envisioned such setups?**
>
> We have not yet conducted a broad investigation across different model architectures, but we acknowledge that **DUO’s effectiveness depends on the system's ability to fully hide its extra full‑precision ReduceScatter within the compute window**.
>
> In particular, DUO may be **less effective** in scenarios where **computation is short relative to communication**, such as:
> - **Very short sequence lengths**
> - **Small gradient accumulation steps**
> - **Severely bandwidth-constrained environments**
>
> In such cases, there may not be sufficient computation time to overlap the added communication, and DUO’s traffic may become visible, diminishing its efficiency. These trade-offs are analyzed in detail in **Appendix D**, which maps out the regimes where overlap begins to break down.
>
> In practice, a straightforward mitigation is to **increase the gradient accumulation step size**, which lengthens the compute window and restores DUO’s ability to overlap and reclaim bandwidth.
>
> We thank the reviewer for raising this point and refer to **Appendix D** for further discussion.

---

> > ### Comment · Reviewer_o9Va · 2025-08-06
> > **Follow up after rebuttal**
> >
> > Thanks to the authors for their detailed clarifications and sharing their more broader vision for this work. For me one of the main points that I find interesting to discuss is situations where DUO can be further improved (small computation scenario as was highlighted). This discussion could strengthen the paper and make the trade offs it more transparent.
> >
> > Overall, I think the work is great and I will raise my score from 5 to 6.

---

### Official Review · Reviewer_vD9N · 2025-07-02

**Clarity:** 3
**Significance:** 4
**Originality:** 3
**Rating:** 5
**Confidence:** 3

**Summary:**

This paper proposes a distributed training framework that alleviates the accuracy loss caused by aggressive gradient compression. By combining AllGather and ReduceScatter and exploiting the idle periods between synchronization steps to overlap communication of additional high-precision gradients, the accuracy degradation due to gradient compression is effectively reduced. Convergence of the proposed algorithm is established, and experiments on GPT models (125M-18B parameters) pretrained on The Pile validate its effectiveness. Overall, the topic is interesting and timely.

**Questions:**

The framework requires saving full-precision gradients, copying them out-of-band (GPU→CPU) via a GPU memory bridge, and scheduling asynchronous reductions, which seems to be fairly complex. Could the authors comment on whether it can be readily integrated into existing codebases?

**Ethical Concerns:**

["NO or VERY MINOR ethics concerns only"]

**Final Justification:**

The authors have addressed my concerns and questions, and I will keep my positive score.

**Limitations:**

The authors have discussed some limitations in the checklist.

**Paper Formatting Concerns:**

There are no formatting concerns.

**Quality:**

3

**Strengths And Weaknesses:**

Strengths:
1. Distributed pre-training of large language models is of paramount interest nowadays, and the compensation for the degradation in accuracy due to gradient compression is an important research direction.

2. Convergence of the proposed algorithm is established from a theoretical perspective, and experiments on various sizes of GPT models are conducted, which are quite comprehensive.

3. Overall, the paper is well-written.

Weaknesses:

I do not identify major weaknesses in this paper. That being said, the proposed strategy depends on having idle compute time between communication phases. In settings with a few micro-batches, there would be little idle time to hide the extra full-precision ReduceScatter. It would be helpful to discuss how DUO performs when accumulation is limited in the main text.

---

> ### Author Rebuttal · Authors · 2025-07-30
>
> # Answer to Weakness
>
> Thank you for highlighting this point. Indeed, a key component of **DUO's system optimization** is the overlapping of **full-precision gradient communication** with the **backward computation phase**. For this to be fully effective, the compute time must exceed the time required for the added communication introduced by the **Fast–Slow algorithm**.
>
> DUO may be **less effective** in scenarios where **computation is short relative to communication**, such as:
> - **Very short sequence lengths**
> - **Small gradient accumulation steps**
> - **Severely bandwidth-constrained environments**
>
> In such settings, there may not be sufficient idle compute time to fully hide the additional communication, and DUO’s overhead may become partially exposed, reducing its efficiency.
>
> We analyze these trade-offs in detail in **Appendix D**, which maps out the regimes where overlap starts to break down. As a **practical remedy**, increasing the **gradient accumulation step size** extends the compute window and restores DUO’s ability to fully overlap communication and reclaim bandwidth.
>
> We will clarify this limitation in the **main text**, and thank the reviewer again for the thoughtful suggestion.
>
> # Answer to Question
>
> Yes, we have already integrated **DUO** into **Megatron-LM**, one of the three most widely used frameworks for large-scale LLM training. Our implementation demonstrates that **DUO can be incorporated with minimal modifications** to the existing training loop, despite involving GPU–CPU transfers and asynchronous scheduling.
>
> We will **open-source the code upon acceptance**, enabling easy adoption and integration by the broader community.

---

> ### Comment · Reviewer_vD9N · 2025-08-02
> **Thanks for your clarification.**
>
> I appreciate the authors for the clarification. I will keep my score.

---

### Official Review · Reviewer_GiNG · 2025-07-02

**Clarity:** 3
**Significance:** 3
**Originality:** 3
**Rating:** 5
**Confidence:** 4

**Summary:**

This paper introduces DUO (Double Update Overlap), a communication-efficient distributed training framework designed to mitigate the accuracy degradation caused by gradient compression in LLM training. The key idea is leveraging otherwise idle communication time to incorporate an additional high-precision gradient communication phase that overlaps with computation. The authors propose the Fast-Slow Algorithm, which mitigates accuracy degradation by incorporating a high-precision gradient to influence optimizer state updates without compromising throughput. This paper provides a theoretical proof of convergence for DUO and validate its effectiveness on GPT-2 model training. Results show that DUO can maintain the accuracy of 4-bit gradient compression to levels comparable with uncompressed training, even with 1-bit gradient compression it still can provide high accuracy.

**Questions:**

Questions:

Typically, gradient compression with biased compressors requires error feedback. How does DUO converge without error feedback? And could the authors provide a more detailed comparison and discussion between DUO and error feedback?
Is DUO also compatible with other compressors, such as top-k sparsification?
The 0-bit (no gradient transmission) is confusing, could the authors provide some more detailed explanation?
Can DUO’s 1-bit (or 0-bit) training maintain its current performance on different datasets and larger-scale models, such as those with 18B models?

**Ethical Concerns:**

["NO or VERY MINOR ethics concerns only"]

**Limitations:**

The authors do not provide a discussion of the limitations of their work; see Weaknesses.

**Quality:**

3

**Strengths And Weaknesses:**

Strengths:

This paper conducts experiments on GPT models with sizes up to 6.7B. DUO achieves accuracy comparable to uncompressed training, even under 1-bit gradient compression or even 0-bit. Notably, in the 1-bit setting, DUO surpasses SDP4Bit in terms of the loss curve while introducing only minimal additional memory overhead. The 0-bit setting is also a very interesting idea.
The paper provides a system-algorithm co-design, leveraging otherwise idle communication time to incorporate high-precision gradients to reduce the communication overhead without compromising the training accuracy.
The integration of algorithmic innovation (Fast-Slow Algorithm) with system optimizations (memory footprint optimization, computation overlapping) makes DUO a practical system for distributed LLM training. The paper also provides substantial implementation details, making the approach more likely to be reproducible.
Theoretical analysis of convergence is also provided for smooth but non-convex settings.

Weaknesses:

The paper focuses on gradient compression, but there is very limited comparison or discussion related to error feedback methods, theoretically and empirically.
The compression techniques used in the experiments are limited to simple quantization.
The 0-bit (no gradient transmission) is confusing, which lacks detailed explanation.
While the end-to-end throughput experiments are scaled up to 18B models, the largest model size for the validation loss experiments is up to only 6.7B, which is relatively small.

---

> ### Author Rebuttal · Authors · 2025-07-30
>
> ### **Weakness**
>
> Since the questions are very similar, we integrate our response directly into the question section.
>
>
> ### **Question - Typically, gradient compression with biased compressors requires error feedback. How does DUO converge without error feedback? And could the authors provide a more detailed comparison and discussion between DUO and error feedback? Is DUO also compatible with other compressors, such as top-k sparsification? The 0-bit (no gradient transmission) is confusing, could the authors provide some more detailed explanation? Can DUO’s 1-bit (or 0-bit) training maintain its current performance on different datasets and larger-scale models, such as those with 18B models?**
>
> DUO employs an alternative approach to error feedback by using a **recursive error correction mechanism** to compensate for errors introduced by gradient and weight-difference compression, rather than relying on traditional error feedback methods explicitly. As detailed in **Theorem 4.1**, DUO recursively corrects errors arising from compressed gradients and weight differences through a **delayed, high-precision gradient synchronization step**. The **theoretical convergence guarantee** provided in **Theorem 4.6** further ensures that DUO converges at a comparable rate to standard SGD, even without traditional error feedback.
>
> Empirically, as shown in **Figure 1**, **Error Feedback performs suboptimally under large-scale training**. In our view, the failure of classic **Error Feedback (EF)** mechanisms under aggressive compression may stem from their **incompatibility with the Adam optimizer and large-scale LLM training**. Prior EF work (e.g., [1] using SignSGD) focused mainly on **SGD-style optimizers** and relatively small CV models, where EF achieved strong results.
> However, a **theoretical analysis directly comparing** the relationship between DUO and error feedback mechanisms remains an **interesting future research direction**.
>
> Regarding compressor compatibility: **DUO is compressor-agnostic**, as long as the compressor is compatible with **Sharded Data Parallelism (SDP)**. This includes quantization methods like **SDP4Bit**, as well as potential compatibility with **sparsity-based approaches** (e.g., Top‑k sparsification), though we leave this as **promising future work**.
>
> We acknowledge that the **0-bit mode** can be confusing and will revise the manuscript to clarify this further. Specifically:
>
> In the **0-bit setting**, DUO transmits **zero bits of gradient data** in the compressed communication channel. This can be viewed as replacing the global (Reduce-Scattered) gradient with the **local pre-communication gradient**. Since **no communication is performed**, this is termed “0-bit.” The optimizer still receives a **meaningful local gradient**—it is not an empty tensor.
>
> In the full DUO configuration, we maintain **two gradient channels** (see **Algorithm 1, Lines 5 and 12**):
> 1) **Timely but compressed communication**
> 2) **Accurate but one-step-stale asynchronous communication**
>
> This co-design delivers **fresh but compressed gradients** at each step and provides **exact correction one step later**. The **0-bit variant** disables the compressed communication channel entirely, relying only on the **delayed accurate channel**. This reduces communication even further—**ideal for highly bandwidth-constrained environments**—but typically comes with a **small drop in final accuracy**.
>
> As for scalability: we have tested DUO across a range of **GPT model sizes**, and we have **not observed any degradation trend** in performance as model size increases. While our largest accuracy experiments are currently up to **6.7B**, we are actively working on extending this to **18B-scale pretraining** and believe DUO will continue to perform well under **larger models and datasets**.

---

### Official Review · Reviewer_Gttj · 2025-07-03

**Clarity:** 3
**Significance:** 1
**Originality:** 1
**Rating:** 2
**Confidence:** 4

**Summary:**

This paper introduces DUO (Double Update Overlap), a distributed training framework that aims to mitigate potential accuracy degradation caused by gradient compression and without incurring additional overhead.

Specifically, DUO inserts an additional high-precision gradient synchronization step during the computation phase, aiming for the communication to be fully hidden by computation.

The paper also provides theoretical proof of convergence for DUO under smoothness and bounded variance assumptions and validates its effectiveness through pre-training experiments on GPT models.

**Questions:**

Can the authors elaborate on the value of compressed versus accurate gradient transmission in DUO? Namely, why is compression even required if the 0-bit approach seems to offer near-baseline accuracy?

**Ethical Concerns:**

["NO or VERY MINOR ethics concerns only"]

**Limitations:**

The paper states that the proposed method is effective only when communication can be overlapped with computation.

**Paper Formatting Concerns:**

N/A.

**Quality:**

2

**Strengths And Weaknesses:**

Pros:

•	Usage of computation-communication overlap to mask communication delays

•	Maintains minimal accuracy degradation under extreme compression (e.g., 1 bit per parameter) in the tested scenarios

•	Optimizes memory usage through high-precision gradient offloading


Cons:

•	Unclear novelty - the idea of using computation phases to mask communication delays is standard practice

•	Related work and evaluation leave more to be desired - numerous strong compression baselines and systems are not considered (e.g., see [1] and references within)

• Increased Communication Overhead - While accurate transmission is achieved during the computation phase, it still increases the communication overhead; thus, the comparison to other compression techniques should take this additional budget into account.

•	The comparison to error feedback is not convincing, and the presented results seem to go against the previous conclusion (e.g., [2]).

[1] Xu, Hang, et al. "Compressed communication for distributed deep learning: Survey and quantitative evaluation." (2020).

[2] Karimireddy, Sai Praneeth, et al. "Error feedback fixes signsgd and other gradient compression schemes." International Conference on Machine Learning. PMLR, 2019.

---

> ### Author Rebuttal · Authors · 2025-07-30
>
> ### **C1 - Unclear novelty - the idea of using computation phases to mask communication delays is standard practice**
>
> Our proposed **DUO** is a **system–algorithm co-design**, and this integration is central to our contribution—it cannot be decomposed into independent components. The co-design consists of two parts:
> (1) A novel algorithm (Sec. 3.1) called **Fast–Slow**, which mitigates compression-induced errors; and
> (2) A **system-level optimization** (Sec. 3.2–3.3) that enables effective overlap of communication introduced by this algorithm.
>
> The idea of using computation to mask communication is indeed well-known, but in our work, it is not a generic overlap strategy—it is **specifically tailored to the Fast–Slow algorithm**, and their **combination** forms the core of DUO’s contribution. More importantly, our overlapping mechanism is fundamentally different from conventional strategies used in existing frameworks.
>
> As explained in **Sec. 2.3** and **Figure 2**, while frameworks like **ZeRO**, **Megatron‑LM**, and **FSDP** implement conventional overlap (e.g., pre‑AllGather + post‑ReduceScatter), they only hide communication for the **final micro-batch** in each gradient accumulation window (that is 1/n of total backward computation). When accumulation has **n** steps, these methods overlap at most **1/n** of the backward pass, leaving the remaining **(n – 1)/n** exposed to communication delays.
>
> In contrast, our **Fast–Slow algorithm** introduces a new **high-precision synchronization step** that can be **delayed to the next iteration**. This allows DUO to overlap communication with the **(n – 1)/n** portion of the backward pass that conventional frameworks leave uncovered. As a result, **DUO significantly improves overlap efficiency** beyond what existing strategies achieve.
>
> ---
>
> ### **C2 - Related work and evaluation leave more to be desired - numerous strong compression baselines and systems are not considered (e.g., see [1] and references within)**
>
> **DUO** is designed specifically for **Sharded Data Parallel (SDP)** training—used in frameworks such as **DeepSpeed**, **Megatron‑LM**, and **PyTorch‑FSDP**—which have become the standard for large-scale LLM training. To date, only a few works have focused on **communication compression in SDP settings**, namely **ZeRO++** [1], **QSDP** [2], and **SDP4Bit** [3].
>
> In contrast, many methods cited in the referenced survey (e.g., [4]) are evaluated under **naive DDP with SGD optimizers**. These settings are not aligned with current **LLM pretraining practices**, which rely on **SDP and AdamW**. As such, applying those baselines would not lead to a fair or meaningful comparison in our context.
>
> That said, we appreciate the reviewer’s suggestion and will include a discussion of these additional methods in the **Related Work** section to clarify how **DUO** differs.
>
> Finally, we emphasize that our focus is **not on proposing a new compressor**, but rather on **enhancing existing ones via error compensation**. Therefore, a broad empirical study across all compressors is beyond the scope of this work.
>
> [1] ZeRO++: Extremely Efficient Collective Communication for Giant Model Training
> [2] Quantized Distributed Training of Large Models with Convergence Guarantees
> [3] SDP4Bit: Toward 4-bit Communication Quantization in Sharded Data Parallelism for LLM Training
> [4] Compressed Communication for Distributed Deep Learning: Survey and Quantitative Evaluation
>
> ---
>
> ### **C3 - Increased Communication Overhead - While accurate transmission is achieved during the computation phase, it still increases the communication overhead; thus, the comparison to other compression techniques should take this additional budget into account.**
>
> We appreciate the reviewer’s concern about communication overhead. However, **DUO is a system–algorithm co-design**, and its communication should not be considered in isolation. As detailed in Sec. 2.3 and Sec. 3.2, the extra communication introduced by DUO is **fully hidden using our custom overlapping strategy**, which leverages **computation phases that conventional methods leave idle**. As a result, **the end-to-end training time remains unchanged**.
>
> Specifically, DUO introduces one full‑precision gradient ReduceScatter per iteration (equal in size to the model), but this is **carefully overlapped with the earlier stages of the next iteration’s backward computation**. As shown in Figure 2 and Table 7 (Appendix D), this communication fits entirely within the previously unutilized **(n−1)/n portion** of the computation window, which standard methods cannot overlap.
>
> In practical large-scale training, what ultimately matters is **wall-clock throughput and final model quality**, not raw byte count. In modern clusters, **network bandwidth is typically provisioned to remain idle if not fully used**. DUO effectively converts this **unused bandwidth into a “free” accuracy gain**, delivering better convergence without increasing total training time.
>
> ---
>
> ### **C4 - The comparison to error feedback is not convincing, and the presented results seem to go against the previous conclusion (e.g., [2]).**
>
> We believe the apparent contradiction arises from a difference in **training regimes and model type/scale**. The method in [1] is based on **SignSGD**, which combines SGD with 1-bit compression. This setup differs fundamentally from our setting, which uses **Adam optimizers for large-scale LLM pretraining**.
>
> Moreover, the experiments in [1] are conducted on **small-scale computer vision models**, using a different model architecture, and dataset. The empirical results from [1]—which are valid in their domain—**may not generalize to large language models** trained with **AdamW optimizer**.
>
> As such, **the conclusions drawn from [1] are not transfer meaningfully to our context**.
>
> [1] Karimireddy, Sai Praneeth, et al. "Error feedback fixes signsgd and other gradient compression schemes." International Conference on Machine Learning. PMLR, 2019.
>
>
> ---
>
> ### **Q1 - Can the authors elaborate on the value of compressed versus accurate gradient transmission in DUO? Namely, why is compression even required if the 0-bit approach seems to offer near-baseline accuracy?**
>
> For the 0-bit situation, it actually means **0 bits for data transmission**. This compression setting can be seen as **compressing the global gradients** (obtained after reduce-scatter) into the **local gradients before reduce-scatter**. In ordinary DDP training, the global gradients are the result of applying all-reduce to the local gradients. Since we use local gradients directly without performing reduce-scatter—thus **no data transmission**—this is referred to as “0-bit.” The low-precision optimizer uses these local gradients, **not some 0-bit empty tensor**.
>
> To answer the question of why compression is still required, we provide a more detailed explanation of DUO’s **two types of gradient communication**. In the full DUO setting, we maintain two complementary gradient channels, as shown in **Algorithm 1 (Lines 5 and 12)**:
> 1) **Timely but compressed communication**
> 2) **Accurate but one‑step‑stale asynchronous communication**
>
> This co-design allows DUO to deliver **fresh gradients each step** via the compressed channel, and then **correct them exactly one step later** via the accurate channel. This leads to performance **very close to full-precision training** (as shown in **Table 2**).
>
> The **0‑bit variant** disables the compressed communication channel entirely and relies solely on the **delayed accurate channel**. This reduces communication cost further—making it suitable for **extremely bandwidth-constrained environments**—but typically comes with a **small drop in final accuracy** compared to full DUO.
>
> In summary, **communication for the compressed channel is still valuable when bandwidth allows**, as it improves convergence. The **0‑bit mode** is a **fallback for more extreme settings**.

---

> > ### Comment · Reviewer_Gttj · 2025-08-03
> >
> > I thank the authors for the detailed response and am keeping my score.

---

> ### Author Response · Authors · 2025-08-04
> **Thank you for your response. Are there any remaining concerns?**
>
> We thank the reviewer for the response. Is there any remaining concern or point that would benefit from further clarification?We would be happy to address them.

---

### Official Review · Reviewer_vxA1 · 2025-07-10

**Clarity:** 4
**Significance:** 2
**Originality:** 2
**Rating:** 4
**Confidence:** 4

**Summary:**

Large scale distributed training uses two techniques to overcome communication bottlenecks: gradient compression and overlapping the computation and communication. However, aggresively employing them simulatenously (like the prior SDP4bit algorithm) can potentially lead to degradation of accuracy of the trained model. This paper modifies SDP4bit with an additional "slow" background communication step where a more high precision synchronization happens, relying on potentially unused bandwidth in the network.

**Questions:**

1. What are the current SOTA large scale training methods?
2. How does DUO compare against them empirically?

**Ethical Concerns:**

["NO or VERY MINOR ethics concerns only"]

**Final Justification:**

The reviewer responses have sufficiently addressed my concerns.

**Limitations:**

yes

**Quality:**

2

**Strengths And Weaknesses:**

Strengths

- The paper is easy to read and well written.
- The modification to SDP4bit is well motivated and the experiment setup comparing to SDP4bit is quite thorough. The improvement results are convincing as well and are performed on realistic large scale training.

Wekanesses

- This paer lacks comparision to any other method apart from SDP4bit. It justifies this by claiming SDP4bit is the SOTA, but I do not see evidence for that claim. Even now well establised methods like [PowerSGD](https://arxiv.org/abs/1905.13727) as used to train [DALLE2](https://proceedings.mlr.press/v139/ramesh21a.html) and [CocktailSGD](https://proceedings.mlr.press/v202/wang23t.html) have not been compared, let alone the current SOTA methods.
- The related work discussion is also very sparse - there is no discussion of recent papers on large scale training e.g. [DiLoCo and variants](https://arxiv.org/abs/2502.12996).

---

> ### Author Rebuttal · Authors · 2025-07-30
>
> ### **W1 - This paer lacks comparision to any other method apart from SDP4bit. It justifies this by claiming SDP4bit is the SOTA, but I do not see evidence for that claim. Even now well establised methods like PowerSGD as used to train DALLE2 and CocktailSGD have not been compared, let alone the current SOTA methods.**
>
> We respectfully clarify that SDP4Bit is currently the strongest publicly available compressor designed specifically for Sharded Data Parallelism (SDP)—a paradigm widely adopted in large-scale LLM training frameworks such as Megatron-LM, DeepSpeed-ZeRO, and PyTorch-FSDP.
>
> Within the scope of communication compression for **Sharded Data Parallelism (SDP)**, only a few peer-reviewed methods exist—namely, ZeRO++, QSDP, and SDP4Bit. Among these, SDP4Bit has already conducted comprehensive comparisons against ZeRO++ and QSDP and has demonstrated superior performance. Based on this, we selected SDP4Bit as our primary baselines.
>
> Our proposed **DUO** is a **system–algorithm co-design**, whose core contribution is to **compensate the accuracy loss of any compressor** under SDP. It is compatible with **arbitrary compressors**, unlike ZeRO++/QSDP/SDP4Bit which rely on unbiased compression. Thus, DUO is orthogonal to the choice of compressor, and the focus of our paper is not to benchmark compressors per se, but to demonstrate how **DUO improves training quality**.
>
> Regarding PowerSGD, it is not compatible with SDP. PowerSGD assumes layer-wise AllReduce of full gradients, while Megatron-LM SDP uses flattened, sharded gradients with ReduceScatter, making such operations infeasible. Notably, DALLE-2 used PowerSGD only for inter-replica communication in **DDP**, not in the context of **sharded training**.
>
> As for **CocktailSGD**, it is optimized for **extremely low-bandwidth fine-tuning scenarios**, with compression ratios exceeding 100×, and has **not been validated for accuracy-sensitive, large-scale pretraining**.
> Moreover, Its experiments are conducted under DDP and pipeline parallelism, not Sharded Data Parallelism (SDP). Its use of **Top‑K sparsification** is **not directly compatible with ReduceScatter-based communication**, and may introduce **additional challenges for system-level optimization**.
>
> ---
>
> ### **W2 - The related work discussion is also very sparse - there is no discussion of recent papers on large scale training e.g. DiLoCo and variants.**
>
> We thank the reviewer’s advice and will expand our **Related Work** section accordingly to clarify this distinction and cite **DiLoCo** and related variants.
>
> In particular, **DiLoCo** and similar **LocalSGD-based methods** are **orthogonal to DUO**. The combination of **Distributed Data Parallelism (DDP)** and **Sharded Data Parallelism (SDP)** has already been explored in prior work, such as **HSDP** [1]. **DUO and DiLoCo** can, in fact, be combined in **HSDP-style setups** to further reduce communication overhead. In this setting:
> - **DiLoCo** reduces **inter-replica communication** by lowering synchronization frequency
> - **DUO** reduces **intra-replica communication** through error-compensated compression
>
> To elaborate: **LocalSGD-style methods** aim to reduce communication by allowing several local update steps before synchronizing. These typically assume each worker holds the entire model (or identical copies), which differs from **SDP**, where model parameters and optimizer states are **sharded across workers** and communication relies on **ReduceScatter/AllGather**.
>
> [1] HSDP: Accelerating Large-scale Model Training via Efficient Sharded Data Parallelism
>
> ---
>
> ### **Q1 - What are the current SOTA large scale training methods?**
>
> We assume the “**training methods**” referred to here concerns **training frameworks**. Widely adopted systems include **Megatron-LM**, **DeepSpeed-ZeRO**, and **PyTorch-FSDP**, all of which implement **Sharded Data Parallelism (SDP)**. SDP is one of the most effective and widely used mechanisms for large-scale model training. It works by **partitioning the optimizer states**—often the most memory-intensive component of model training—so that each worker holds only a **shard** of the states and updates its corresponding parameters. Synchronization across workers is performed via **ReduceScatter** and **AllGather** operations.
>
> For **communication compression tailored to SDP**, the main methods are **ZeRO++**, **QSDP**, and **SDP4Bit**. Among them, **SDP4Bit** has already provided thorough comparisons against ZeRO++ and QSDP, and currently stands as the **strongest open-source baseline** for gradient compression under SDP.
>
> Other compression techniques such as **PowerSGD**, **QSGD**, and **SignSGD** are mostly designed and evaluated under **naive DDP settings**. Most of them lack validation in **large-scale LLM pretraining tasks**, particularly when used with **Adam optimizers** and **Sharded Data Parallel (SDP)** setups, making them less applicable to our setting. Though PowerSGD is not compatible with SDP, QSGD and SignSGD are compatible. Note that the combination of QSGD or SignSGD with SDP is conceptually the same as ZeRO++ or QSDP.
>
> ---
>
> ### **Q2 - How does DUO compare against them empirically?**
>
> As explained in Q1, SDP‑4Bit is already a strong representative of current SOTA compressors for SDP. We choose to apply DUO on top of SDP4Bit, thereby demonstrating how DUO can improve final model accuracy while maintaining its high compression ratio.
> For the compressors, we choose the quantization as our compression methods to keep our evaluation consistent to the previous works such as QSDP and ZeRO++. Note that the focus of our paper is not to benchmark compressors.

---

> > ### Comment · Reviewer_vxA1 · 2025-08-01
> >
> > Thank you for your clarification regarding related work. This indeed adresses my concerns. However, I enourage the authors to describe in more detail how their setup (SDP) differs from standard DPP setup and the unqiue challenges this setup brings about in more detail in their final version.

---

### Note · Authors · 2025-08-12

We thank the reviewers for their careful evaluation and constructive feedback.

During the rebuttal, we provided clarifications and additional results addressing: (i) our choice of SDP4Bit as the SOTA baseline under Sharded Data Parallelism and the fairness of this comparison; (ii) DUO’s core contribution (the Fast–Slow algorithm and its system-level optimizations) and its distinctions from generic compute–communication overlap strategies; (iii) how the added full-precision communication is budgeted and, in common regimes, hidden by backward compute so end-to-end throughput is preserved; (iv) the relationship to Error Feedback (EF), noting differences in experimental setup—Sharded Data Parallelism with Adam and ReduceScatter/AllGather—that limit direct apples-to-apples comparisons; (v) limitations and trade-offs, notably regimes with short sequences, few micro-batches, or small gradient-accumulation steps where overlap may be incomplete; and (vi) DUO’s 0-bit gradient transmission. We appreciate that reviewers confirmed their concerns were addressed and maintained or increased their scores.

Regarding Reviewer Gttj: We appreciate the reviewer’s summary and questions. In our rebuttal, we addressed the points on novelty, baseline coverage, communication budgeting, and the differences with Error Feedback (EF), and we provided a detailed explanation of DUO’s 0-bit transmission mode and the complementary roles of its two gradient communication channels—a timely, compressed path and a delayed, accurate correction path. As of this writing, we have not seen further comments from the reviewer; we remain happy to clarify any remaining points.

If accepted, we will integrate these clarifications into the main text, and improve presentation for readability.

---

### Decision · Program_Chairs · 2025-09-17

**Decision:**

Accept (poster)

**Comment:**

To overlap compute and communication in large-scale and distributed settings, the authors have proposed to leverage idle synchronization intervals to mask communication of high-precision gradients and compensate for the performance drop due to compression.

The paper was thoroughly discussed by our reviewers and has mixed reviews. Most reviewers are already happy with the paper and rebuttal. Reviewer Gttj is mainly concerned regarding the novelty of making communication delays and missing major compression baselines.

The authors' rebuttal has further clarified the key novelty compared to pre‑AllGather and post‑ReduceScatter methods in terms of overlap ratio of backward pass.

Overall, I think this is a good paper. I would suggest that authors address all revisons that are promised within the camera-ready version.